# Chromatin accessibility dynamics dictate renal tubular epithelial cell response to injury

Xinyi Cao [1,2], Jiuchen Wang[1,3], Tianye Zhang[1], Zhiheng Liu [1], Lijun Liu[4], Ying Chen[4], Zehua Li[4], Youlu Zhao[4], Qi Yu[4], Tong Liu[2], Jing Nie[5], Yuanjie Niu[6], Yupeng Chen [1] ✉, Li Yang [4] ✉ & Lirong Zhang [1] ✉

Renal tubular epithelial cells (TECs) can initiate an adaptive response to completely recover from mild acute kidney injury (AKI), whereas severe injury often leads to persistence of maladaptive repair and progression to kidney fibrosis. Through profiling of active DNA regulatory elements by ATAC-seq, we reveal widespread, dynamic changes in the chromatin accessibility of TECs after ischemia–reperfusion injury. We show that injury-specific domains of regulatory chromatin become accessible prior to gene activation, creating poised chromatin states to activate the consequent gene expression program and injury response. We further identify RXRα as a key transcription factor in promoting adaptive repair. Activation of RXRα by bexarotene, an FDA-approved RXRα agonist, restores the chromatin state and gene expression program to protect TECs against severe kidney injury. Together, our findings elucidate a chromatin-mediated mechanism underlying differential responses of TECs to varying injuries and identify RXRα as a therapeutic target of acute kidney injury.

Acute kidney injury (AKI) is a prevalent clinical syndrome, manifesting as a sudden decline of renal function that occurs over hours or days, which leads to the accumulation of waste products and damage to kidneys and the rest of the body. A variety of conditions can cause AKI, such as renal ischemia, sepsis, exposure to nephrotoxic drugs, and infections[1–3]. AKI accounts for about 2 million deaths worldwide every year, and about 30–70% of AKI survivors develop chronic kidney disease (CKD) and eventually end-stage renal disease[1,4]. At present, AKI patients can only be treated by supportive therapies, and effective therapies to reduce tissue damage or promote repair are still lacking.

Renal tubular epithelial cells (TECs) are known to be most vulnerable to AKI[5]. After mild injury, mature surviving TECs can initiate an adaptive repair process, undergoing dedifferentiation, proliferation, and subsequent redifferentiation to replenish lost epithelial cells[6,7]. In contrast, TECs often undergo a maladaptive response following severe injury, leading to a chronically injured and profibrotic phenotype[8–10].

[1]Key Laboratory of Immune Microenvironment and Disease (Ministry of Education), The Province and Ministry Co-sponsored Collaborative Innovation Center for Medical Epigenetics, Department of Biochemistry and Molecular Biology, School of Basic Medical Sciences, Tianjin Institute of Urology, The Second Hospital of Tianjin Medical University, Tianjin Medical University, Tianjin, China. [2]Tianjin Key Laboratory of Ionic-Molecular Function of Cardiovascular Disease, Department of Cardiology, Tianjin Institute of Cardiology, The Second Hospital of Tianjin Medical University, Tianjin, China. [3]Department of Reproductive Medical Center, Taihe Hospital, Hubei University of Medicine, Shiyan, China. [4]Renal Division, Peking University First Hospital; Institute of Nephrology, Peking University, Key Laboratory of Renal Disease, Ministry of Health of China, Key Laboratory of CKD Prevention and Treatment (Peking University), Ministry of Education of China, Beijing, China. [5]State Key Laboratory of Organ Failure Research, National Clinical Research Center of Kidney Disease, Key Laboratory of Organ Failure Research (Ministry of Education), Division of Nephrology, Nanfang Hospital, Southern Medical University, Guangzhou, China. [6]Department of Urology, The Second Hospital of Tianjin Medical University, Tianjin Medical University, Tianjin, China. ✉e-mail: ychen@tmu.edu.cn; li.yang@bjmu.edu.cn; lzhang@tmu.edu.cn

Recently, transcriptomic studies from bulk, cell-specific, and single-cell analyses have revealed new molecular pathways and gene regulatory networks that underlie successful and failed renal epithelial cell repair[11–17]. However, the fundamental molecular mechanisms and key factors in the transcriptional responses and fate decisions of TECs following kidney injuries of different severity are still elusive.

Accumulating evidence indicates that epigenetic regulation plays critical roles in cell fate decision by governing temporal and cell type-specific gene expression programs[18,19]. Chromatin accessibility, referring to the degree of physical compaction of chromatin, is a key characteristic of chromatin states and is generally recognized as a common property of active *cis*-regulatory elements[20–22]. With fewer nucleosomes and less chromatin compaction, accessible chromatin regions are where transcription factors (TFs) are recruited via DNA-specific interactions. Meanwhile, the binding of TFs also contributes to establishing and maintaining the openness of these regulatory regions. According to the sequence characteristics of accessible chromatin regions, the binding of TFs can be predicted and dynamic TF regulatory networks can be mapped[20]. Therefore, characterizing chromatin dynamics and profiling TF regulatory networks after AKI may elucidate the epigenetic underpinnings of cellular responses to varying degrees of injury and thus identify key regulators in adaptive and maladaptive kidney repair.

In this study, we profiled the dynamic changes in chromatin accessibility and gene expression, and constructed and distinguished the TF regulatory networks, in mouse renal TECs after AKI of different severity. We identified the nuclear receptor RXRα as a major TF that determined the differential chromatin states and gene expression program in response to mild or severe AKI, and we provide evidence that activation of RXRα protects TECs from severe AKI.

## Results

### Characterization of chromatin dynamics of TECs after AKI

To decipher the epigenetic mechanisms underlying the distinct responses of TECs to various severities of AKI, we first set up two AKI mouse models: a mild ischemia–reperfusion injury (MI) model with 20-min bilateral renal ischemia and a severe ischemia–reperfusion injury (SI) model with 30-min bilateral ischemia. Mice were euthanized at three time points (2, 7, and 30 days) after AKI and compared with mice receiving sham surgery (Supplementary Fig. 1a). Assessment of serum creatinine (Scr) and blood urea nitrogen (BUN) indicated a rapid induction of renal dysfunction in MI and SI mice (Supplementary Fig. 1b). In addition, the mRNA levels of two kidney injury markers, kidney injury molecule-1 (KIM-1, encoded by *Havcr1*) and neutrophil gelatinase-associated lipocalin (NGAL, encoded by *Lcn2*), were markedly increased in whole kidney tissue and isolated TECs (Supplementary Fig. 1c and d). An increase of KIM-1 staining was observed in TECs two days after MI and SI (Supplementary Fig. 1e). By 30 days after injury, the mRNA levels of two fibrogenic factors, *Col3a1* and *Fn1*, were higher in SI mice than in MI and sham groups (Supplementary Fig. 1f). In addition, Masson's trichrome staining, and immunohistochemistry staining for α-smooth muscle actin (α-SMA), fibronectin, and collagen-1, revealed substantial tubulointerstitial fibrosis in SI mice compared to MI mice and sham-operated mice (Supplementary Fig. 1g). Collectively, these results demonstrate that MI can resolve with adaptive repair to functional recovery, while SI induces maladaptive repair leading to fibrosis.

To explore genome-wide chromatin changes during adaptive and maladaptive repair after AKI, we examined temporal chromatin accessibility dynamics with ATAC-seq (assay for transposase-accessible chromatin by sequencing) in TECs following sham surgery, MI, or SI (Figs. 1a and 2a). To minimize survival bias and increase data consistency, we chose mice with serum creatinine in the range of $2.12 \pm 0.17$ mg/dL in SI group and of $1.3 \pm 0.24$ mg/dL in MI group for the subsequent sequencing analysis (Supplementary Fig. 2). The purity

of isolated TECs was measured by LTL immunofluorescence staining analysis. As shown in Supplementary Fig. 3a, LTL-positive cells exceeded 90% for each condition and no differences were observed among sham, MI, and SI groups. Inflammatory cells infiltrate after injury, especially in kidneys after SI[23]. We measured the expression of an immune cell marker gene (CD45, encoded by *Ptprc*), a macrophage marker gene (*Cd68*), and a T-cell marker gene (*Dd3e*) in the whole cortex and purified tubule fragments. The expression of these immune cell marker genes was markedly increased after injury in whole cortex, with the highest expression in the SI group. In contrast, in purified tubular cells, the expression levels of these marker genes were very low and did not differ between the injured and control groups (Supplementary Fig. 3b). These results suggest that tubular cells were isolated with high purity. All ATAC-seq libraries were sequenced to around 50 million reads to ensure sufficient coverage across the genome. ATAC-seq data were highly reproducible between two biological replicates (Supplementary Fig. 4a and b), and we therefore merged the data from two replicates at each time point for subsequent analysis.

We first applied a soft clustering approach to separate differential chromatin accessibility regions (DARs) in TECs from MI and sham surgery samples. We identified four clusters of genomic regions with unique temporal dynamics of chromatin openness. As shown in Fig. 1b, 1020 peaks (Sham-DARs) displayed decreased accessibility after injury; 1632 (Day 2-DARs) became more accessible on day 2, but thereafter decreased to basal level; 185 (Day 7-DARs) were open at day 7 and reverted to closed by day 30; and 206 peaks (Day 30-DARs) transitioned from a closed to an open chromatin state by day 30. As shown in Fig. 1c, the majority of the DARs were located in promoter and distal regulatory regions. To examine the biological processes related to these DARs, we assigned each DAR peak to the nearest gene and performed gene ontology (GO) analysis for these four gene sets. This revealed that Sham-DAR genomic regions were enriched for genes involved in multiple metabolic processes (e.g., *Nr1h3* and *Pck1*), ion transport (e.g., *Vdac1* and *Slc25a25*), and response to oxygen levels (e.g., *Sod3* and *Cited2*) (Fig. 1d), suggesting that these elements represent the normal epigenomic signature of proximal tubule cells (PTCs) homeostasis. Day 2-DARs genomic sites associated with genes related to cell adhesion and cell migration, such as cell junction assembly (e.g., *Ctnna1*, *Vmp1*, and *Rapgef2*), epithelial cell migration (e.g., *Bcar1*, *Src*, and *Hspb1*), and cell-substrate adhesion (e.g., *Abl1*, *Ctnnb1*, and *Crkl*). Day 7-DARs chromatin regions were enriched for genes related to morphogenesis, such as determination of bilateral symmetry (e.g., *Traf3ip1* and *Notch1*), regulation of cation channel activity (e.g., *Fkbp1a* and *Kcng1*), negative regulation of regulated secretory pathway (e.g., *Abr* and *Rap1b*), and positive regulation of phagocytosis (e.g., *Calr* and *Appl2*), suggesting that genes regulated by these elements promote the recovery of injured TECs. Day 30-DARs genomic sites associated with genes related to carbohydrate metabolic process (e.g., *Rpia* and *Edem1*) and nucleic acid transport (e.g., *Cetn3* and *Rftn1*). Representative track profiles of ATAC-seq are shown in Fig. 1e.

We next performed the same clustering of DARs in TECs after SI (Fig. 2a, b). These DARs also displayed enriched distribution in promoter and distal regulatory genomic regions (Fig. 2c). GO analysis revealed that Sham-DARs genomic regions were enriched for genes involved in multiple metabolic processes, such as fructose 1,6-bisphosphate metabolic process (e.g., *Aldob* and *Fbp2*), fatty acid metabolic process (e.g., *Acnat1/2*, *Cd36*, and *Acsm2*), and alpha-amino acid metabolic process (e.g., *Tpk1*, *Ubiad1*, and *Mtr*) (Fig. 2d), consistent with the high metabolic rate in kidney TECs. Day 2-DARs genomic regions were associated with genes related to actin filament-based process (e.g., *Cx3cl1*, *Arhgef5*, and *Nf2*) and regulation of cellular response to growth factor stimulus (e.g., *Fgfr1*, *Abl1*, and *Ptpn1*), representing an injury response epigenomic signature. Day 7-DARs genomic regions were enriched for genes involved in multiple

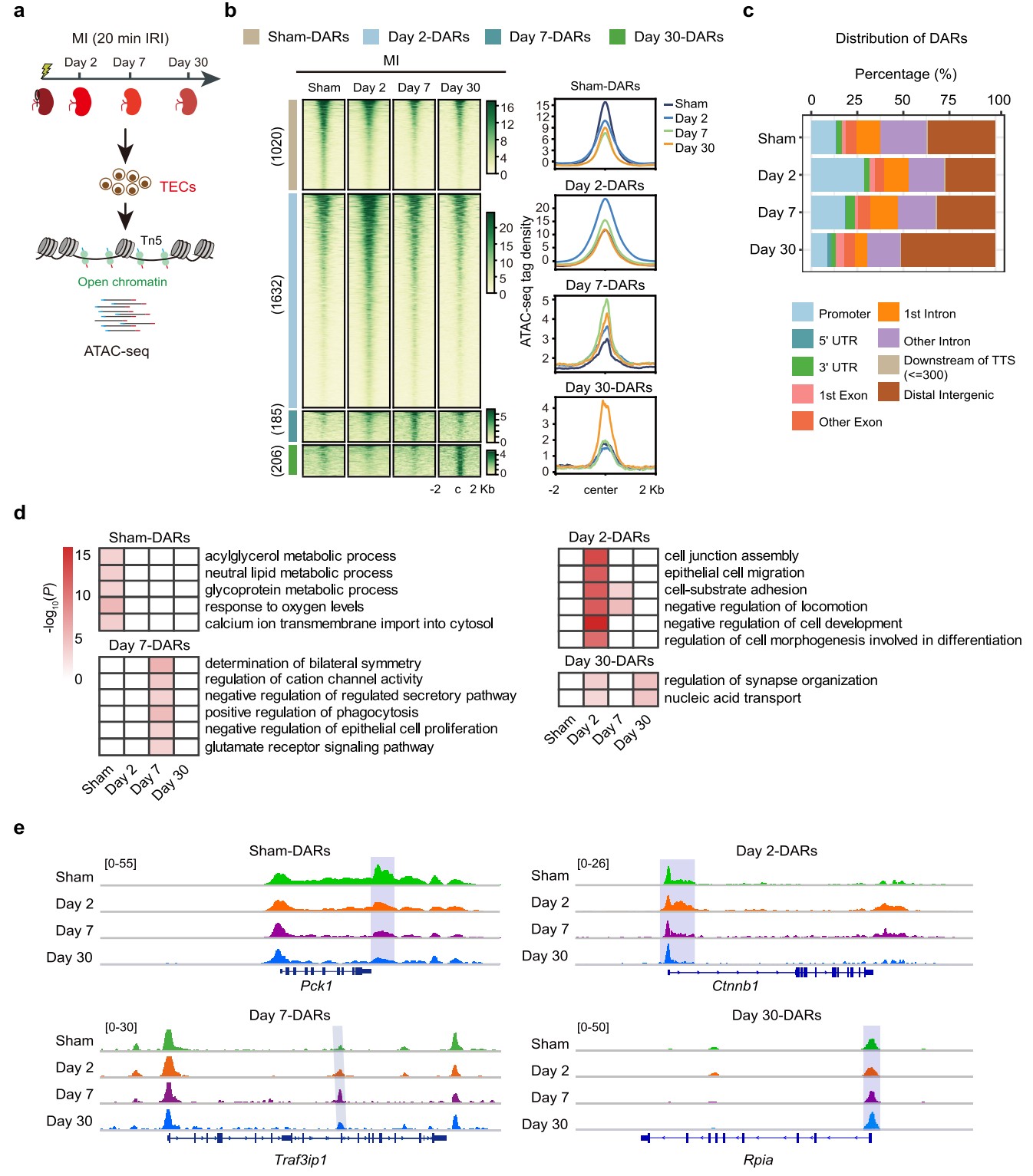

**Fig. 1 | Chromatin accessibility landscape of TECs after mild kidney injury.**
**a** Experimental strategy for genome-wide ATAC-seq assay and time points of TEC collection after mild injury. **b** Temporal changes in chromatin accessibility during adaptive repair. Peaks are ordered vertically by ATAC-seq signal strength. The signal strengths are denoted by color intensities. **c** Genomic distribution of DARs at the indicated time points after mild injury. **d** Heatmaps of Biological Process GO terms in each cluster. *P* values were calculated by clusterProfiler R package and denoted by color intensities. **e** Genome browser view showing representative DARs at the indicated gene loci for TECs after mild injury. Source data are provided as a Source Data file.

inflammatory pathways (e.g., *Cd44*, *Ccl2/3/12*, and *Ccrl2*), indicating an inflammatory response at this time point. Day 30-DARs genomic regions were associated with genes involved in multiple ion transport pathways. Representative track profiles are shown in Fig. 2e.

Collectively, by mapping the active DNA regulatory elements and connecting them to associated genes, these results reveal temporal changes in chromatin accessibility of TECs in response to different injuries.

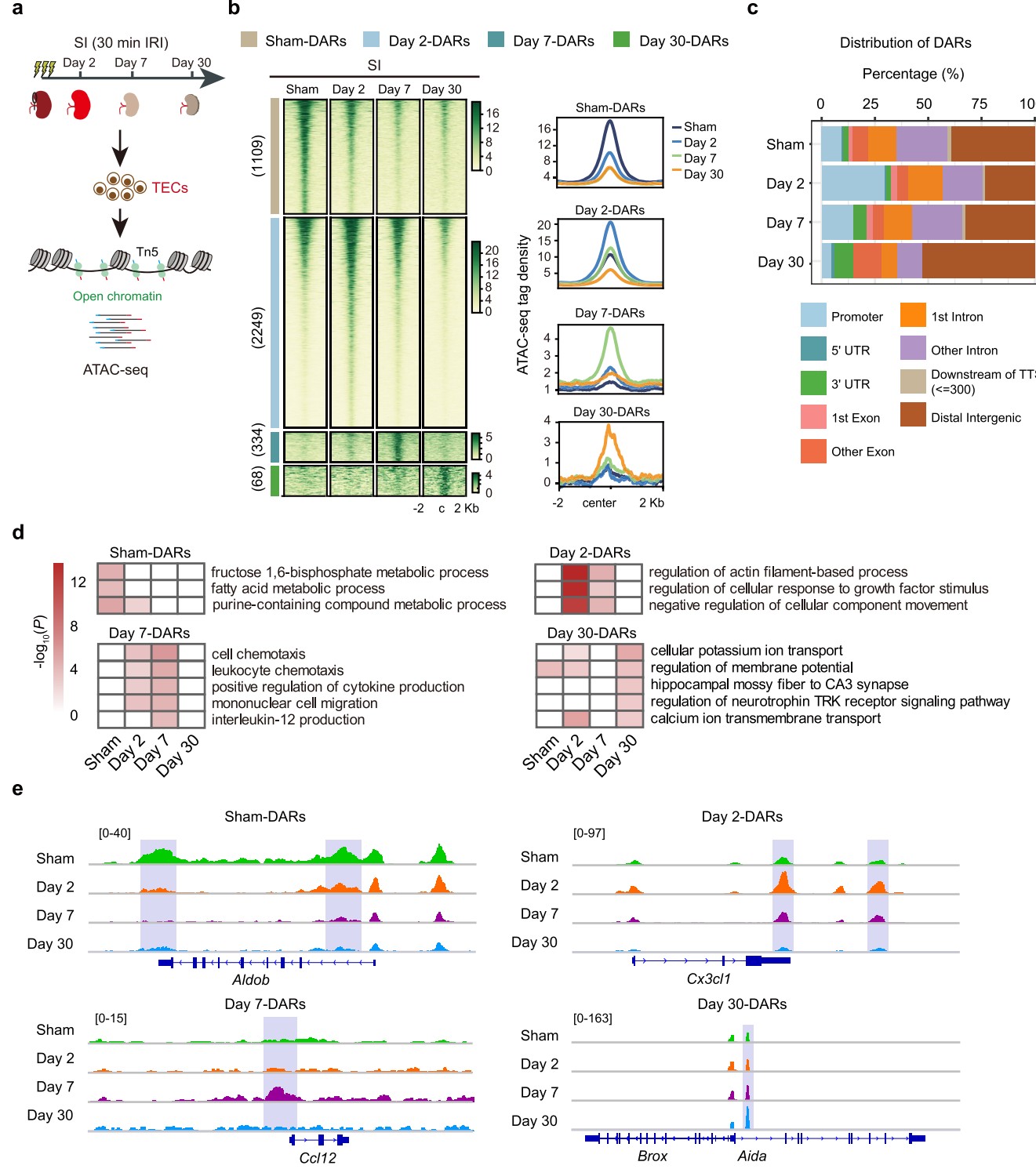

**Fig. 2 | Chromatin accessibility landscape of TECs after severe kidney injury.**
**a** Experimental strategy for genome-wide ATAC-seq assay and time points of TEC collection after severe injury. **b** Temporal changes in chromatin accessibility during maladaptive repair. Peaks are ordered vertically by ATAC-seq signal strength. The signal strengths are denoted by color intensities. **c** Genomic distribution of DARs at the indicated time points after severe injury. **d** Heatmaps of Biological Process GO terms in each cluster. *P* values were calculated by clusterProfiler R package and denoted by color intensities. **e** Genome browser view showing representative DARs at the indicated gene loci for TECs after severe injury. Source data are provided as a Source Data file.

## Injury-specific chromatin openness precedes gene activation

To decipher the chromatin events that distinguish adaptive and maladaptive repair, we performed a comparative analysis on chromatin accessibility profiles of TECs at day 2 and day 7 following MI and SI. Interrogation of ATAC-seq data from day 2 after AKI identified 4172

peaks with higher accessibility in MI TECs (MI-DARs) and 4253 peaks with higher accessibility in SI TECs (SI-DARs) (Fig. 3a, left panel). At day 7 after AKI, 6309 and 2616 peaks were identified as MI-DARs and SI-DARs, respectively (Fig. 3a, right panel). Thus, in total there were 8425 and 8925 DARs between MI and SI on day 2 and day 7, respectively. As

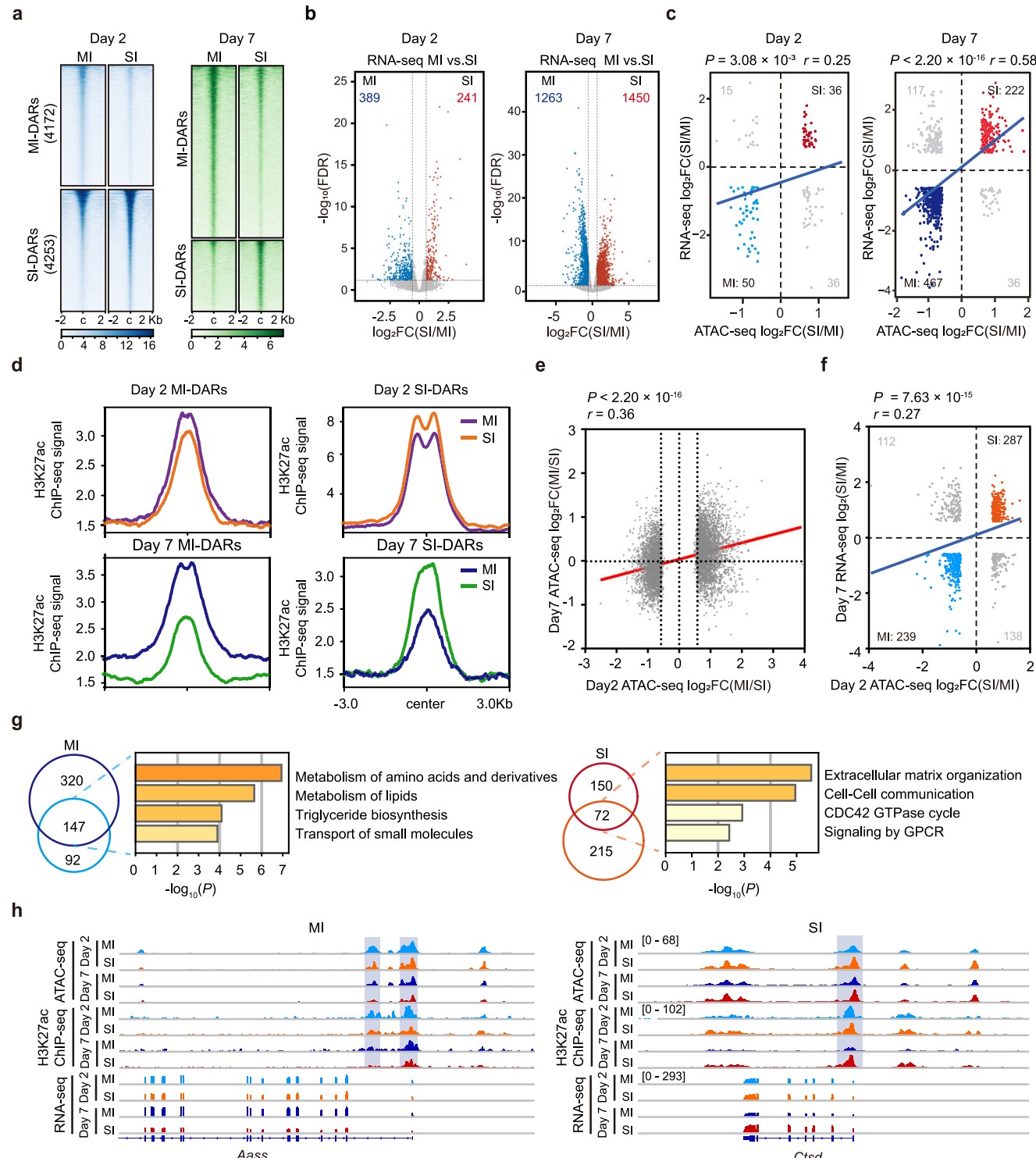

**Fig. 3 | Injury-specific chromatin opening dictates gene activation. a** Heatmap visualization of ATAC-seq signals in Sham, MI, and SI TECs centered on peak summits ± 2 Kb. Peaks are ordered vertically by ATAC-seq signal strength. The signal strength are denoted by color intensities. **b** Volcano plots showing differentially expressed genes (DEGs) (MI versus SI). **c** Scatterplots of the DEGs and DARs-associated genes. Red indicates genes that are more accessible and upregulated in SI, and blue indicates genes that are more accessible and upregulated in MI. For Day 2, $P = 3.08 \times 10^{-3}$; for Day 7, $P < 2.20 \times 10^{-16}$. **d** Visualization of H3K27ac ChIP-seq signals in Sham, MI, and SI TECs centered on peak summits ± 3 Kb. **e** Scatterplot of

the fold change (MI versus SI) of ATAC-seq signals in TECs at day 2 and day 7. $P < 2.20 \times 10^{-16}$. **f** Scatterplot of day 2 DARs-associated genes and day 7 DEGs. $P = 7.63 \times 10^{-15}$. **g** Pathway analysis of DEGs within day 2 DARs and day 7 DARs in mild or severe injury. Venn diagram shows overlap of MI- and SI-DARs-associated genes identified in **c** and **f**. **h** Genome browser view showing DARs at the indicated gene loci for TECs. Red represents increase, and blue represents decrease (**b**, **c**, **f**). Statistical significance was analyzed by cor.test (**c**, **e**, **f**) or Metascape (https://metascape.org/gp/index.html) (**g**). Source data are provided as a Source Data file.

shown in Supplementary Fig. 5, compared with the SI group, the ATAC-seq profiles of MI were more similar to Sham. These results indicate that MI TECs and SI TECs exhibit distinct chromatin accessibility landscapes at both day 2 and day 7 after AKI, and a profound difference in open chromatin domains between MI and SI emerges as early as day 2.

To explore whether the differences in chromatin openness correlate with distinct gene expression in TECs of MI and SI, we then performed RNA sequencing (RNA-seq) in TECs from both AKI models at day 2 and day 7, as well as in TECs from sham-operated mice. We found that 389 and 241 genes were differentially expressed (fold change > 1.5) in TECs from MI and SI at day 2 (Fig. 3b). By contrast, by 7 days after AKI, the number of differentially expressed genes (DEGs) in TECs among different AKI models increased strikingly to 2713 (1263 genes downregulated and 1450 upregulated in SI TECs in comparison to MI TECs) (Fig. 3b). These data indicate that, in contrast to the rapid changes in chromatin openness, the alterations in gene expression are markedly delayed. Meanwhile, integrative analysis of ATAC-seq and RNA-seq data showed that the differences in chromatin accessibility correlated weakly with the changes in gene expression between MI TECs and SI TECs at day 2 after AKI ($P = 3.08 \times 10^{-3}$) (Fig. 3c). In contrast, at day 7 after injury, changes in chromatin accessibility exhibited a strong correlation with changes in gene expression ($P < 2.20 \times 10^{-16}$), with 222 genes exhibiting higher chromatin accessibility and increased gene expression while 467 genes displayed lower chromatin accessibility and decreased gene expression in SI TECs compared to MI TECs (Fig. 3c). H3K27ac is a well-characterized active histone mark. We next performed H3K27ac ChIP-seq analysis in purified TECs after Sham surgery, MI, and SI, and analyzed H3K27ac peak signals in MI- and SI-DARs at day 2 and day 7. Consistent with gene expression, the differential H3K27ac peak signals were higher at day 7 than at day 2 in both MI- and SI-DARs (Fig. 3d).

Interestingly, DARs between MI and SI exhibited a positive correlation between day 2 and day 7 (Fig. 3e), suggesting that these DARs remain accessible over time. Furthermore, integrative analysis of the DARs (MI vs. SI) at day 2 revealed a positive correlation with DEGs (MI vs. SI) at day 7 for a total of 776 genes ($P = 7.63 \times 10^{-15}$) (Fig. 3f). Pathway analysis revealed that regions with lower chromatin accessibility and gene expression in SI TECs were enriched for renal functional genes, such as genes related to the metabolism of amino acids and their derivatives, triglyceride biosynthesis, and transport of small molecules (Fig. 3g, h), indicating a dysfunctional state of SI TECs. By contrast, genes related to pathways such as signaling by extracellular matrix organization and cell–cell communication were overrepresented among genes with higher chromatin accessibility and gene expression in SI TECs (Fig. 3g, h), implying a higher fibrotic and inflammatory potential of SI TECs. Collectively, the above results demonstrate, when comparing the response of TECs to MI and SI, that chromatin domains are accessible prior to gene expression, and that the early chromatin events foreshadow the later gene expression program.

## DARs are enriched for RXRα motifs during adaptive repair

Open chromatin domains are largely established and maintained by TFs to control gene expression. To identify the key TFs that are responsible for modulating the dynamics of chromatin states in response to different kidney injuries, we analyzed TF motifs on MI- and SI-DARs. As shown in Fig. 4a, MI-DARs were enriched for DNA-binding motifs of TFs associated with renal development and metabolism, such as nuclear receptor (NR), homeobox, and zinc finger (ZF) proteins (upper panel). Conversely, these chromatin sites in SI-DARs were mainly enriched for motifs of stress-related TFs, including bZIP, TEAD, and ETS TFs (lower panel).

Quantification of coverage of TF-binding motifs showed that six (ERRA, NR2F2, EAR2, PPARA, LHX1, and RXR) and seven (ERG, ETV1,

EHF, ELF4, ETV4, FLI1, and TEAD3) TF motifs were found in over 50% of MI- and SI-DARs, respectively (Fig. 4b, c). TFs often interact to form regulatory networks and thus synergistically enhance transcriptional activity[24,25]. To assess the interactions and cooperation of these TFs, we mapped TF regulatory networks on MI- or SI-DARs at both day 2 and day 7 after injury. We found that HNF4α, PPARα, and RXRα were the core TFs regulating chromatin architecture of MI-DARs (Fig. 4d), while FOSL2 and MEF2A were key TFs enriched in SI-DARs (Fig. 4e). HNF4α has previously been shown to control the expression of PTC-specific transmembrane transport and metabolic genes, and to regulate enhancer dynamics during kidney repair[26]. PPARα has been shown to protect against metabolic and inflammatory derangement in sepsis-associated acute kidney injury[27]. Thus, the discovery of these previously AKI-related TFs verifies the feasibility of our network analysis for identifying TFs that are potentially involved in the injury response. As shown in Fig. 4d, RXRα was the most prominently enriched TF in the network in TECs after MI. Since the mechanism and function of RXRα in ischemia–reperfusion kidney injury has not yet been reported, we therefore focused on RXRα for further analysis.

## RXRα binding correlates with chromatin openness and gene expression

To validate the bioinformatic data, we first profiled the genome-wide occupancy of RXRα through chromatin immunoprecipitation followed by sequencing (ChIP-seq) analysis in TECs after MI. As shown in Fig. 5a, RXRα occupancy was detected in about 25% of MI-DARs. Next, we divided MI-DARs into two groups: RXRα-bound and RXRα-unbound regions. To study whether RXRα occupancy correlates with accessible chromatin, integrative analysis of RXRα ChIP-seq and ATAC-seq profiles were performed. We found that RXRα-bound MI-DARs were more accessible than those without RXRα binding (Fig. 5a, b). Further correlation analysis showed that RXRα binding was positively correlated with chromatin openness (Fig. 5c). We then explored the relationship between RXRα occupancy and gene expression by interrogating RXRα ChIP-seq and RNA-seq profiles of MI TECs. As shown in Fig. 5d, among genes associated with MI-DARs, the mRNA levels of RXRα-bound genes were significantly higher than those of RXRα-unbound genes ($P = 1.76 \times 10^{-13}$). Collectively, these results indicate that RXRα binding correlates positively with both chromatin openness and gene activation in TECs after MI.

Since the RXRα-binding motif was specifically enriched in MI-DARs, we then wondered whether RXRα occupancy in these regions decreased after SI, which might lead to the failure of these regions to open and thus of activation of associated genes in SI TECs. To address this, we first examined RXRα expression. As shown in Fig. 5e, f, mRNA and protein expression of RXRα in TECs both decreased after SI compared to MI and Sham surgery groups, which explains the defects of RXRα binding following SI. Consistent with this, RXRα ChIP-seq analysis revealed that both RXRα binding and chromatin accessibility in MI-DARs markedly decreased in SI TECs (Fig. 5g). To further explore the impact of decreased RXRα binding on gene expression in SI TECs, we compared gene expression of these RXRα-bound and MI-DARs-associated genes (a total of 1194 genes) between MI and SI TECs. Among these 1194 genes, expression of 319 genes was lower and of 36 genes was higher in SI TECs compared to MI TECs (Fig. 5h). GO analysis of the 319 upregulated RXRα target genes revealed an enrichment of pathways related to multiple metabolic processes and ion transportation (Fig. 5h), which are critical for renal function and repair recovery. We therefore defined these 319 genes as RXRα-activated genes. Representative track files for RXRα ChIP-seq, ATAC-seq, and RNA-seq are shown in Fig. 5i. Taken together, these findings indicate that RXRα binding correlates with chromatin accessibility and gene activation in MI TECs, and that loss of RXRα recruitment might underlie the

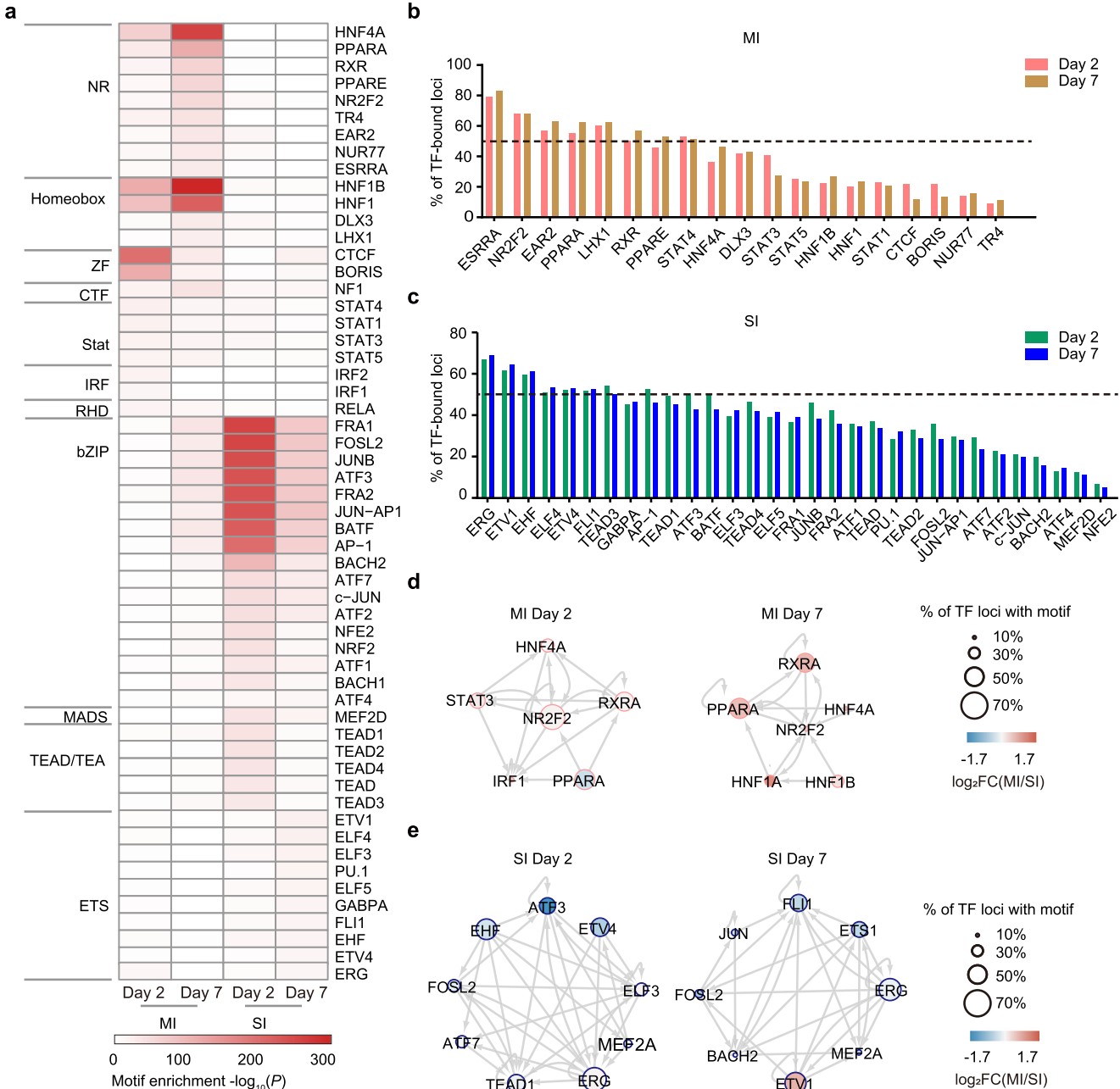

**Fig. 4 | Transcription factor regulatory networks in adaptive and maladaptive repair. a** TF motif enrichment in DARs shown in Fig. 3a. *P* values were calculated by HOMER, using the binomial distribution. The color bar indicates Log10(*P*). **b, c** Quantification of coverage of TF-binding motifs. **d, e** TF regulatory networks for day 2 (left) and day 7 (right). Node color represents TF expression and node size represents percentage of predicted TF-binding sites. The diameter of the dot on the right indicates the proportion of TF loci within the motif. The color bars on the left indicate the foldchange of each gene. Red represents an increase, and blue represents decrease. Source data are provided as a Source Data file.

defects in chromatin openness, gene expression, and tubule recovery after SI.

### Activation of RXRα protects TECs against SI

Next, we asked whether activating RXRα can promote renal repair and ameliorate renal fibrosis after SI. Bexarotene (Bex) is a third-generation retinoid and an FDA-approved drug that selectively activates RXRs. We assessed the effect of pretreatment with Bex in SI mice. As shown in Fig. 6a, Bex administration greatly decreased the mortality of SI mice, with survival rate increasing from around 41.66 to 100% by day 7 after SI (Fig. 6a). Bex treatment also reduced tubular injury (Fig. 6b) and serum creatinine level (Fig. 6c) at day 2 after SI. By day 30, we observed

a substantial decrease of BUN levels in mice treated with Bex (Fig. 6d), indicating an improved renal function. Measurement of mRNA levels of *Col3a1* and *Fn1* (Fig. 6e), assessment of renal fibrosis by Masson's trichrome staining, and quantification of α-SMA-positive areas (Fig. 6f) revealed a marked decrease of fibrosis in kidneys treated with Bex. Collectively, these findings indicate that activation of RXRα protects TECs against severe kidney injury.

### Bex restores chromatin state and gene expression program after SI

We next sought to elucidate the molecular mechanisms underlying Bex-mediated renal protection against SI. Since profoundly differential

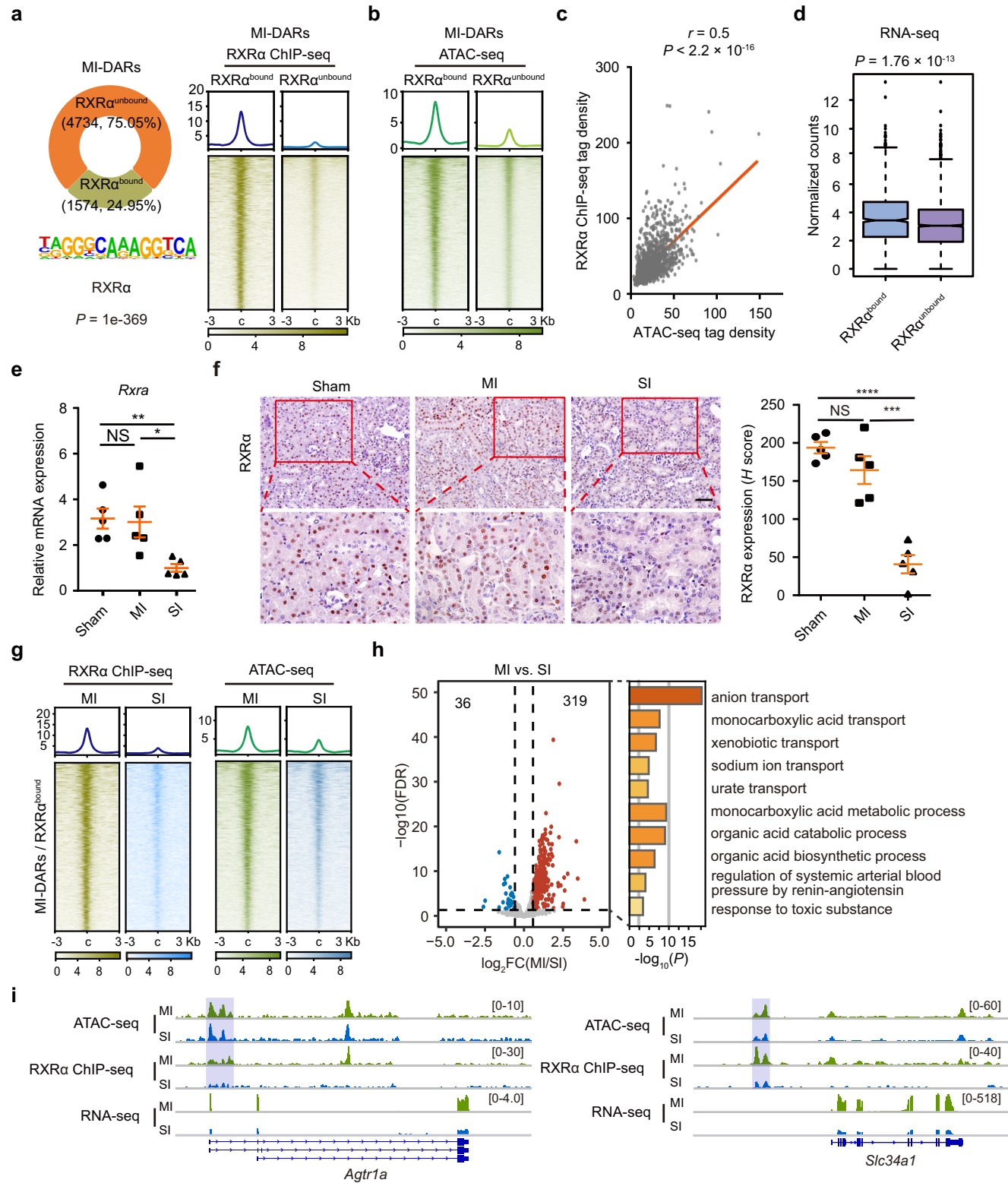

RXRα binding, chromatin accessibility, and gene expression were observed at day 7 after MI and SI, we first evaluated the impact of Bex treatment on RXRα occupancy on chromatin at day 7. We performed RXRα ChIP-seq analysis in SI mice treated with Bex, and compared it with RXRα ChIP-seq profiles in Sham surgery, MI, and SI without Bex treatment. As shown in Fig. 7a, the genomic occupancy of RXRα in MI-DARs was markedly increased upon Bex treatment in SI mice, restoring RXRα binding to a level comparable to that in MI mice. Similarly, ATAC-seq analyses revealed a reprogramming of chromatin accessibility in

MI-DARs in SI mice after Bex treatment (Fig. 7b). In addition, the increase of RXRα binding at MI-DARs correlated positively with the increase of chromatin accessibility in SI mice upon Bex treatment (Fig. 7c). Importantly, Bex treatment induced a marked upregulation of RXRα-activated genes (Fig. 7d, e). Representative track profiles of RXRα ChIP-seq, ATAC-seq, and RNA-seq are shown in Fig. 7f. Taken together, these findings suggest that Bex treatment activates genes by increasing RXRα recruitment and RXRα-associated chromatin remodeling, and thus protects TECs from acute injury in SI mice.

**Fig. 5 | Binding of RXRα correlates with chromatin openness and gene expression. a** Occupancy of RXRα on MI-DARs. The percentage of RXRα-bound sites and RXRα motif enrichment of MI-DARs (left). *P* value was calculated by HOMER, using the binomial distribution. ATAC-seq signals in DARs (right). **b** ATAC-seq signals on RXRα^bound and RXRα^unbound sites. **c** Pearson's correlation coefficients of RXRα ChIP-seq signals and ATAC-seq signals on RXRα^bound regions. Statistical significance was determined by the Correlation test. $P < 2.2 \times 10^{-16}$. **d** Gene expression levels of RXRα^bound and RXRα^unbound site-associated genes. Box plots represent median values, 25%, and 75% quantiles. Whiskers extend to 1.5 times the interquartile range. Statistical significance was determined by the two-sided Mann–Whitney *U* test. $P = 1.76 \times 10^{-13}$ (RXRα^bound: *n* = 1194 genes, RXRα^unbound: *n* = 2870 genes). **e** qRT-PCR analysis of *Rxra* mRNA in kidney tissues from Sham and

AKI mice. From left to right: \*\**P* = 0.0017, NS *P* = 0.8569, \**P* = 0.0202, respectively, by two-tailed unpaired Student's *t*-test. NS not significant. *n* = 5 biologically independent samples. **f** Immunohistochemistry staining of RXRα in kidney tissues from Sham and AKI mice. From left to right: \*\*\*\**P* < 0.0001, NS *P* = 0.1729, \*\*\**P* = 0.0005, respectively, by two-tailed unpaired Student's *t*-test. NS not significant. *n* = 5 biologically independent samples. Scale bars, 50 μm. **g** RXRα ChIP-seq and ATAC-seq signals in the SI group. **h** Volcano plot showing gene expression of RXRα-activated renoprotective genes. Blue or red represents a decrease and increase, respectively. Statistical analysis was calculated by Metascape. **i** Representative IGV tracks showing RXRα ChIP-seq, ATAC-seq, and RNA-seq signals of these 319 genes. The signal strength is denoted by color intensities (**a**, **b**, **g**). Data are represented as means ± SEM (**e**, **f**). Source data are provided as a Source Data file.

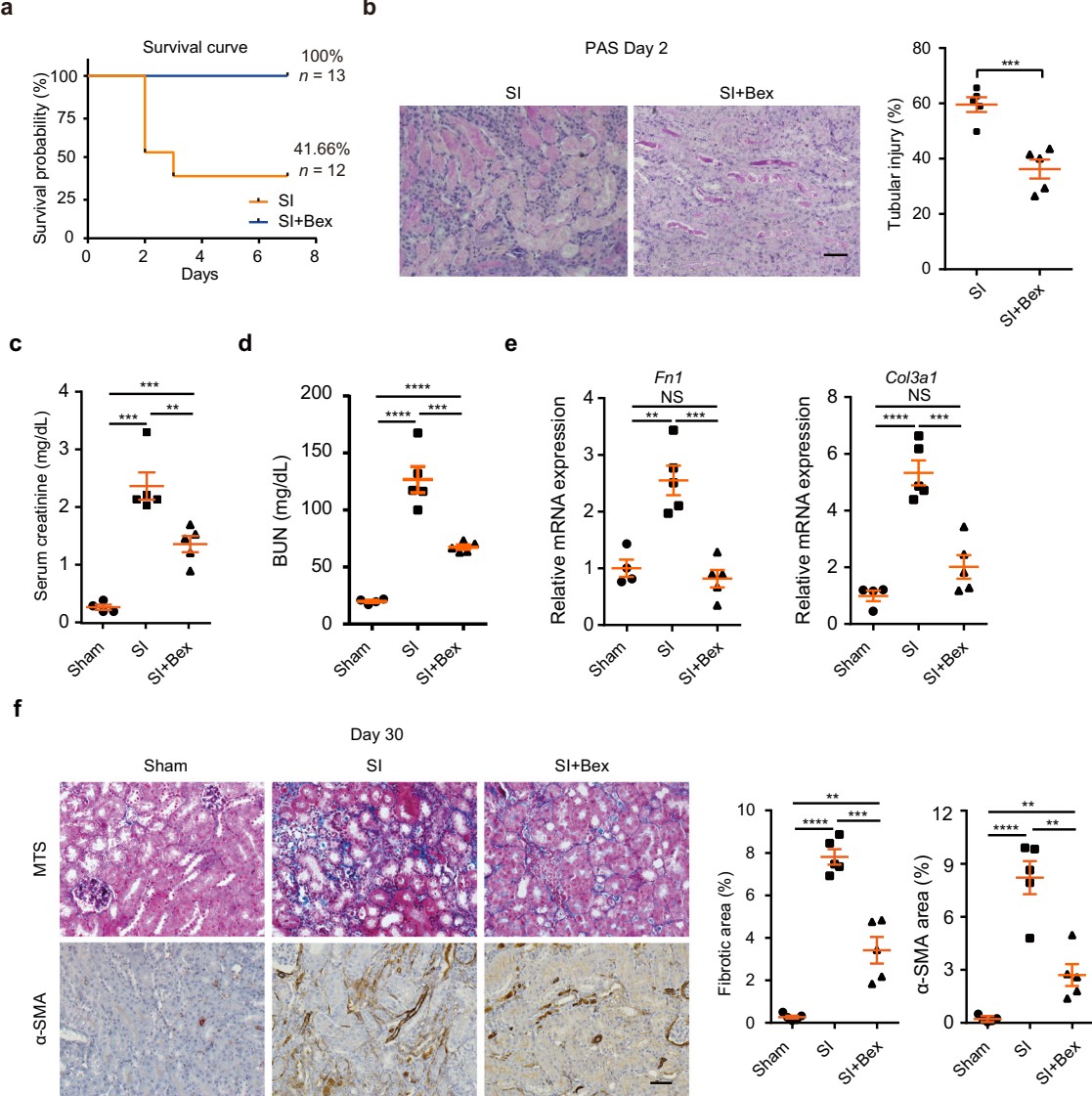

**Fig. 6 | Bex treatment protects TECs against severe injury. a** Kaplan–Meier survival curves for mice with or without Bex treatment. *n* = 13 or 12 biologically independent mice. **b** Representative periodic acid-Schiff (PAS) staining of the kidneys at day 2 after injury (left). Data are expressed as means ± SEM. \*\*\**P* = 0.0007, by two-tailed unpaired Student's *t*-test. *n* = 5 biologically independent mice. Scale bars, 50 μm. Tubular injury scores were analyzed (right). **c** Serum creatinine concentrations in AKI mice with or without Bex treatment at day 2. Data are expressed as means ± SEM and analyzed by two-tailed unpaired Student's *t*-test. From left to right: \*\*\**P* = 0.0003, \*\*\**P* = 0.0001, \*\**P* = 0.0063, respectively. **d** BUN (blood urea nitrogen) concentrations in AKI mice with or without Bex treatment at day 30. Data are expressed as means ± SEM and analyzed by two-tailed unpaired Student's *t*-test. From left to right: \*\*\*\**P* < 0.0001, \*\*\*\*<0.0001, \*\*\**P* = 0.0009,

respectively. **e** qRT-PCR analysis of *Fn1* and *Col3a1* expression. Data are expressed as means ± SEM and analyzed by two-tailed unpaired Student's *t*-test. From left to right for *Fn*: NS *P* = 0.4304, \*\**P* = 0.002, \*\*\**P* = 0.0004, respectively. From left to right for *Col3a1*: NS *P* = 0.08, \*\*\*\**P* < 0.0001, \*\*\**P* = 0.0006, respectively. NS not significant. **f** Masson's trichrome (MTS) staining of kidneys after Bex treatment (upper). Immunofluorescence staining of α-SMA in the kidneys of Bex-treated AKI mouse kidneys (lower). Data are expressed as means ± SEM and analyzed by two-tailed unpaired Student's *t*-test. From left to right for MTS: \*\**P* = 0.0011, \*\*\*\**P* < 0.0001, \*\*\**P* = 0.0003, respectively. From left to right for α-SMA: \*\**P* < 0.0041, \*\*\*\**P* < 0.0001, \*\**P* = 0.0012, respectively. Scale bars, 50 μm. *n* = 4 or 5 biologically independent mice for Sham group (**c**–**f**). *n* = 5 biologically independent mice for SI and SI + Bex groups (**c**–**f**). Source data are provided as a Source Data file.

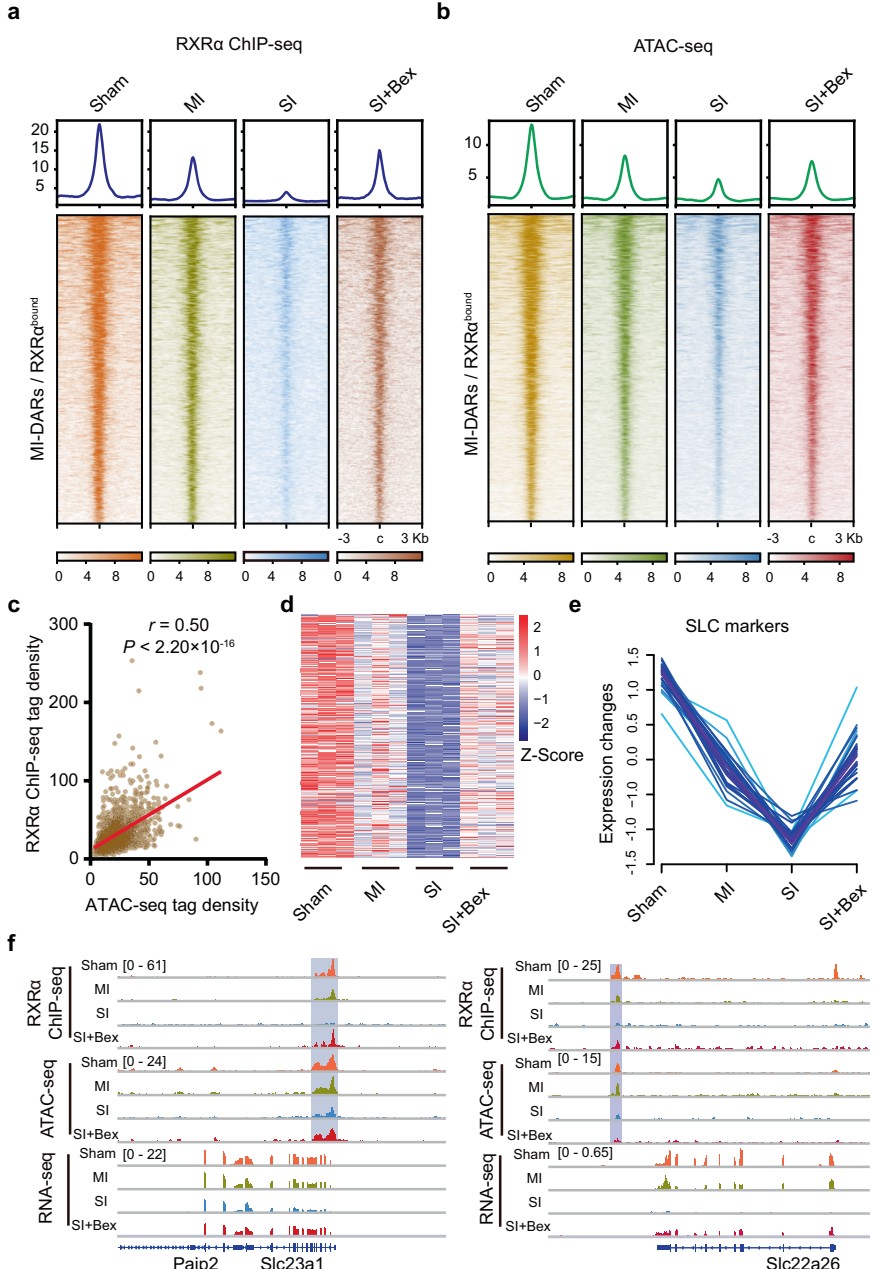

**Fig. 7 | Activation of RXRα reprograms chromatin states and gene expression.**
**a** Occupancy of RXRα on MI-DARs/RXRα$^{bound}$ regions in TECs treated with Bex.
**b** ATAC-seq signals on MI-DARs/RXRα$^{bound}$ regions in TECs treated with Bex.
**c** Pearson's correlation coefficients of RXRα ChIP-seq signals and ATAC-seq signals on MI-DARs/RXRα$^{bound}$ regions in TECs treated with Bex. Statistical significance was determined by the Correlation test. $P < 2.2 \times 10^{-16}$. **d** Gene expression levels of

RXRα-activated renoprotective genes in TECs treated with Bex. The color intensities indicate the Z-score of each gene. **e** Transcript expression levels of SLC markers. **f** Representative IGV tracks showing RXRα ChIP-seq, ATAC-seq, and RNA-seq signals in TECs treated with Bex. The signal strengths are denoted by color intensities (**a**, **b**). Source data are provided as a Source Data file.

## Expression of RXRα correlates with disease severity of AKI patients

Recent single-cell RNA-sequencing (scRNA-seq) analyses have provided further evidence that PTCs are the key vulnerable cell type and revealed the diversity of PTC states in response to AKI[14–17]. To confirm our findings at the single-cell transcriptomic level, we first assessed the expression of RXRα and RXRα-activated genes in PTC subclusters from scRNA-seq datasets of mouse and human AKI[16]. As shown in Fig. 8a–c, the expression of RXRα and RXRα-activated genes decreased with the severity of the injuries. Trajectory analysis showed a positive correlation between kidney injury severity and downregulation of RXRα or of its target genes

that are involved in anion transport and monocarboxylic acid metabolic process (Fig. 8d, e). Next, we analyzed an AKI patient cohort with biopsy-proven acute tubular injury ($n = 28$) to further validate the clinical relevance of RXRα in AKI. The causes of AKI were defined as nephrotoxicity in ten cases, ischemic injury in seven cases, ischemic and nephrotoxic injury in three cases, others in three cases, and unknown in five cases. The clinical manifestations of these AKI patients are listed in Supplementary Table 1. RXRα expression was examined using immunohistochemistry staining on the biopsied sections of AKI patients; para-carcinoma kidney tissues were used as controls. As shown in Fig. 8f, RXRα expression decreased substantially in kidneys from patients with

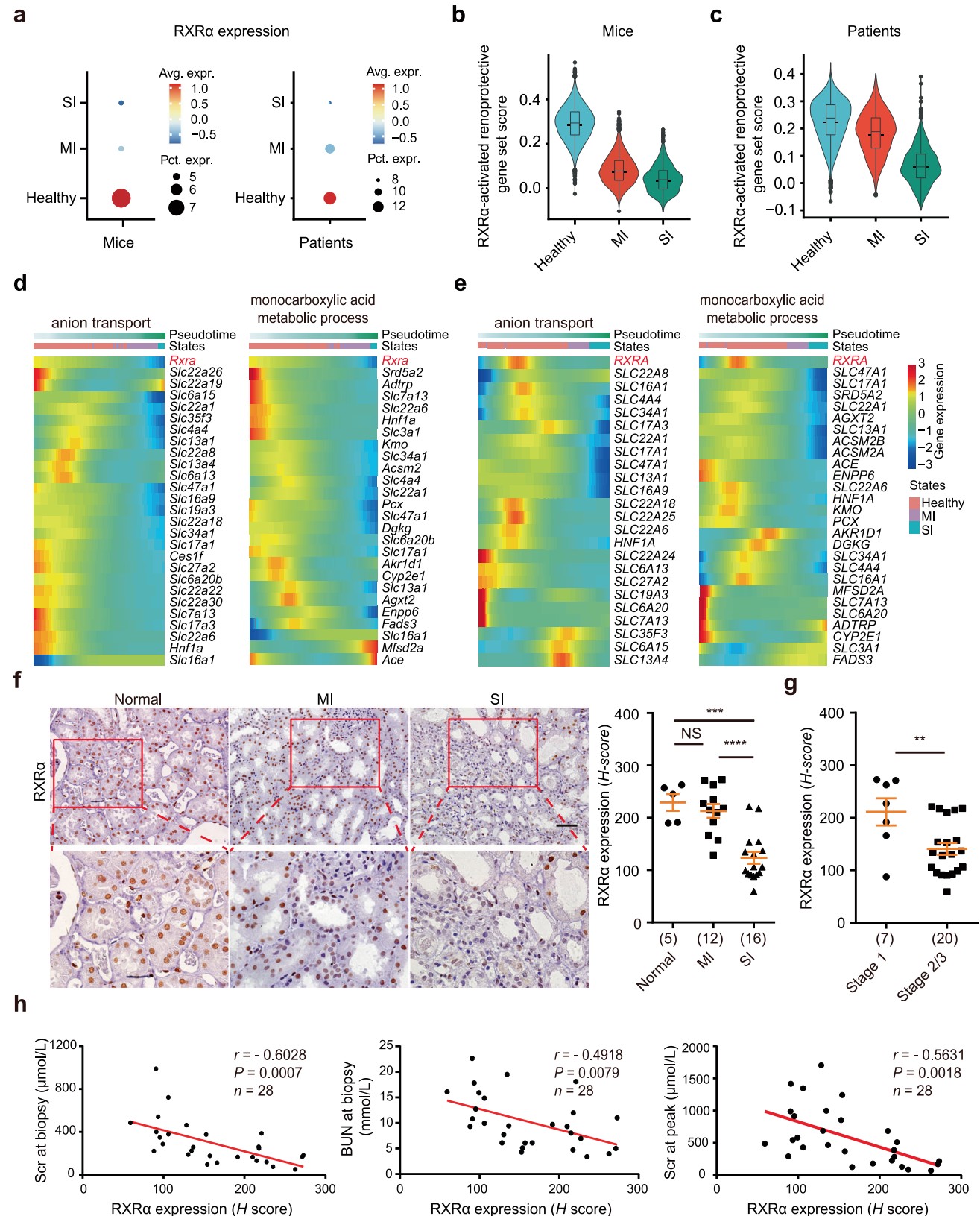

severe AKI. Furthermore, RXRα expression was lower in patients at AKI stages 2 and 3 than in patients at AKI stage 1, indicating that RXRα expression is inversely correlated with the clinical severity of AKI (Fig. 8g). Further analysis showed that RXRα expression level was inversely correlated with the Scr at peak, BUN at peak, and Scr at biopsy (Fig. 8h). These data indicate that renal tubular

RXRα expression is diminished in severe AKI patients, which correlates with more advanced tubular injury and failed renal repair.

## Discussion

Elucidation of the molecular mechanisms underlying repair and regenerative responses after kidney injury is crucial for addressing the

**Fig. 8 | RXRα expression is inversely correlated with kidney disease severity in AKI mice and patients. a** Bubble plot of RXRα expression levels from AKI mice and patients. The color intensities indicate average expression. The diameter of the dot corresponds to the proportion of cells expressing RXRα. Avg. expr. average expression, Pct. expr. percent expressing. **b** Violin plot showing the expression level of RXRα-activated renoprotective genes from Sham and AKI mice. Box plots represent median values, 25%, and 75% quantiles. Whiskers extend to 1.5 times the interquartile range. $n = 29619$ cells (Healthy), $n = 12241$ cells (MI), $n = 1938$ cells (SI). **c** Violin plot showing the expression levels of RXRα-activated renoprotective genes from normal and AKI patients. Box plots represent median values, 25%, and 75% quantiles. Whiskers extend to 1.5 times the interquartile range. $n = 14083$ cells (Healthy), $n = 2195$ cells (MI), $n = 1766$ cells (SI). **d, e** Heatmaps depicting relative expression of the highly variable anion transport (left) and monocarboxylic acid metabolic (right) genes along the mouse (**d**) and human (**e**) PTC injury trajectory.

Heatmap colors represent gene-wise normalized expression across pseudotime. The color bars under the pseudotime are used to separate PTC subclusters. **f** Immunohistochemistry staining of RXRα in kidney tissues from normal and AKI patients (left). Scale bars, 50 μm. Quantification of RXRα expression (right). Data are represented as means ± SEM. From left to right: ***$P = 0.0001$, NS $P = 0.4894$, ****$P < 0.0001$, respectively, by two-tailed unpaired Student's $t$-test. $n = 5$ normal samples, $n = 12$ MI samples, $n = 16$ SI samples. Each dot indicates a biological replicate. **g** Quantification of RXRα expression in patients with different stages of AKI. Data are represented as means ± SEM and were analyzed by two-tailed unpaired Student's $t$-test. **$P = 0.0079$. $n = 7$ for stage 1, $n = 20$ for stage 2/3. **h** Correlation between RXRα expression and kidney fuction. Serum creatinine (Scr) at biopsy (left), BUN at biopsy (middle), and Scr at peak (right) in AKI patients. Pearson's correlation coefficients are displayed. Statistical significance was determined by cor.test. Source data are provided as a Source Data file.

unmet medical needs for AKI treatment. Unfortunately, despite great efforts, a fundamental question remains unanswered, namely, why some TECs can initiate an adaptive response and recover completely, whereas others fail to launch this repair process, which culminates in long-term adverse effects[28]. In this study, we identify chromatin dynamics as a critical mechanism underlying TECs' response to different severities of injury. Through mapping the active DNA regulatory elements, we demonstrate a widespread and dynamic chromatin remodeling over the time course of kidney injury. Furthermore, in a comparison of mild and severe injury, substantial differences in the chromatin accessibility profiles emerge prior to the divergence of gene expression. This finding indicates that differences in chromatin openness are potential early drivers dictating the distinct outcomes for TECs following varying injuries. Importantly, we show that early changes in chromatin openness prime the later gene expression program and injury response. This discovery suggests that early intervention targeting chromatin dynamics represents a potential treatment strategy for AKI.

By analyzing TF regulatory networks, we identify RXRα as a key regulator in modulating accessible chromatin during adaptive repair. Consistent with our study, Lidberg et al. recently showed that RXRα motifs are enriched in open chromatin regions in normal human kidney cortex, but are diminished after injury[29]. Indeed, bexarotene-induced activation of RXRα restores chromatin states, reprograms gene expression, and protects TECs from severe injury. Bexarotene is an FDA-approved drug that has been used orally to treat skin manifestations of cutaneous T-cell lymphoma for more than two decades, without serious toxic or side effects being found in previous animal and human studies[30]. However, evidence that bexarotene protects against early kidney injury directly through activation of RXRα is still lacking. In addition, whether other cellular processes and survival factors are involved in the renal protective effect of bexarotene remains to be explored. Tubular cell-specific RXRα knockout mice will help to answer these questions in future studies; meanwhile, the present study suggests that a chromatin-based mechanism could be targeted as a therapeutic regimen and for drug repurposing in AKI treatment. Accumulating evidence indicates that epigenetic regulation, including DNA methylation, histone modifications, and various non-coding RNAs, plays key roles in the process of kidney repair after injury[31]. Our current study establishes a critical role for chromatin dynamics in injury responses, thus expanding our understanding of the epigenetic mechanisms in AKI and kidney repair.

Dynamic changes in chromatin accessibility control the packaging and expression of a large number of genes, and thereby affect various signaling pathways[20,22]. Rather than regulating a single gene target or cellular pathway, remodeling chromatin can simultaneously modulate multiple injury-associated gene expression programs and signaling pathways, thereby achieving a systemic effect. Genome-wide, chromatin openness and its associated gene expression are tightly controlled by TF regulatory networks. Our TF regulatory network analysis

uncovers the involvement of several key TFs in establishing and maintaining accessible chromatin regions in response to AKI. Besides RXRα, we find that HNFs and PPARα are also highly enriched in the TF network involved in adaptive repair, both of which have previously been shown to be involved in kidney repair[16,26,27]. We also find that the binding motifs of these two TFs overlap considerably with those of RXRα. Integrated with gene expression analysis, these interacting TFs are shown to regulate each other's expression, suggesting that they form a regulatory circuit and cooperate to orchestrate target gene expression. Thus, from a therapeutic perspective, a combination therapy strategy of activating these TFs simultaneously may synergize to improve renoprotective effects[32]. Furthermore, we have also discovered a series of stress-related TFs, including bZIP, TEAD, and ETS, that may function as regulators controlling the openness of chromatin during severe kidney injury[33–35]. Analogous to the therapeutic effects of RXRα activation, modulation of these TFs may regulate chromatin accessibility in SI-specific regions and the expression of SI-associated genes, and thereby ameliorate severe injury and subsequent fibrosis. Overall, our work proposes a therapeutic strategy for the prevention and treatment of AKI through a chromatin-mediated mechanism. The TFs identified in this study warrant further research to understand the progression after AKI and to develop safer and more effective treatment strategies.

## Methods
### Study approval
The study was designed and conducted in accordance with Chinese law and the criteria set by the Declaration of Helsinki. The protocol concerning the use of patient samples in this study was approved by the Biomedical Research Ethics Committee of Peking University First Hospital (approval number: 2017[1280]), and informed consent was obtained from all participants. All mouse experiments were approved by the Ethical Committee of Tianjin Medical University (permit no. SYXK: 2016-0012). KPMP approved the KPMP data usage in our study. The results of human AKI scRNA-seq in this study are based partly on data generated by KPMP: DK114886, DK114861, DK114866, DK114870, DK114908, DK114915, DK114926, DK114907, DK114920, DK114923, DK114933, and DK114937. Data were downloaded on 7 January 2022.

### Patient selection of renal biopsy-AKI retrospective cohort
Twenty-eight patients in the Peking University First Hospital from 2007 to 2020 who were diagnosed with renal tubular acute injury (ATI) or acute tubular necrosis (ATN) were recruited in the study. The median age of the 28 included subjects was 48 years with a range from 15 to 82 years. The degree of renal tubular acute injury was assessed by two renal pathologists in a blinded manner. A 0 to 4+ scale scoring system was used based on the percentage of renal tubules affected by the loss of tubule epithelial cell brush border and tubular necrosis and/or apoptosis (0 = no lesion, 1+ = ≤25%, 2+ = >25 to 50%, 3+ = >50 to 75%, 4+ = >75 to <100%)[36]. Scores of 1 and 2 were defined as mild ATI, and

scores of 3 and 4 as severe ATI. Five para-carcinoma kidney tissues pathologically identified as the healthy parts of the kidney were used as controls. Those who had ATI/ATN concomitant with glomerular or vascular lesions were excluded. Clinical characteristics and pathological evaluations of the enrolled patients are summarized in Supplementary Table 1.

AKI was defined according to the Kidney Disease Improving Global Outcomes (KDIGO) criteria (https://kdigo.org/guidelines/acute-kidney-injury/). We defined the baseline Scr as follows: for patients with a previous Scr in the 365 days prior to admission, the lowest Scr value prior to and during hospitalization was considered the baseline Scr; for patients without a Scr in the 365 days prior to admission, with renal pathology excluding chronic lesions, the baseline Scr was imputed on the basis of a Modification of Diet in Renal Disease (MDRD) eGFR of 75 ml/min per 1.73 m² as per the KDIGO AKI guidelines. AKI stages were defined using the KDIGO AKI stage Scr definitions. AKD and its stages was defined by the Acute Disease Quality Initiative (ADQI) consensus as AKI stage 1 or greater (as defined by KDIGO) that is present between 7 and 90 days after an AKI episode[37]. Peak AKI/AKD stages was defined as the highest stages during hospitalization. Peak Scr was defined as the highest Scr after an AKI/AKD episode until discharge. We defined renal recovery as Scr decreased to the baseline Scr or decreased by 25% or more from peak Scr. And failure to recover was defined as patient still dependent on dialysis or Scr decreased by <25% from peak concentration[38].

### Animal models and drug treatment

Eight-week-old C57BL/6 male mice were purchased from Beijing Vitalstar Biotechnology (Beijing, China). Mice were housed on a 12 h light-dark cycle at 21–25 °C with 30–70% humidity and allowed free access to food and water except as noted. Mice were randomly assigned to experimental groups. For the bexarotene experiment, eight-week-old mice were orally gavaged daily with 100 mg/kg/day of bexarotene (S2098, Selleck) or 10% DMSO/90% corn oil vehicle three days before the procedure. The mice continued to be treated with bexarotene daily for two days after injury.

### Ischemic AKI induction

Kidney bilateral ischemia–reperfusion injury (bIRI) surgery was performed as described[39]. In detail, eight-week-old C57BL/6 male mice were anesthetized by isoflurane inhalation and kept on a heating blanket with body temperature maintained at 36.5 °C. Both kidney pedicles were exposed by the retroperitoneal approach and clamped with microvessel clips for 20 min to generate a mild IRI model and for 30 min to generate a severe IRI model. Sham operations were performed using the same procedure but without induction of ischemia. For volume supplement, 1 ml of warmed saline was injected intraperitoneally after surgery. Mice were killed at 2, 7, or 30 days after surgery.

### Measurement of renal function

Mouse blood samples collected from heart puncture at sacrifice or from the tail vein were clotted and centrifuged to collect serum. Serum creatinine concentration was measured by the picric acid method with a commercial kit (DICT-500, BioAssay Systems), following the manufacturer's instructions. BUN was measured with a commercial kit (DIUR-500, BioAssay Systems) following the manufacturer's instructions.

### Renal histology

For histology, renal specimens were fixed in formalin overnight, and embedded in paraffin. Four-micrometer sections were used for periodic acid-Schiff, Masson's trichrome staining (MTS), immunohistochemistry, and immunofluorescence staining. Tubular injury was defined as brush border loss, necrosis, intratubular debris, cast formation, and atrophy. Tubular injury degree was scored by the percentage of injured tubules in cortex in a blinded manner. The grading percentage in each field was calculated as follows: injury score (%) = (number of injured tubules/number of whole tubules) × 100%. The degree of interstitial fibrosis was assessed with MTS-stained kidney sections using Image-Pro Plus 6.0 software.

### Immunohistochemistry and immunofluorescence

Immunohistochemistry was performed on formalin-fixed, paraffin-embedded kidney sections from human and mouse using antibodies against RXRα (ET7108-99, Huabio, 1:200 dilution), α-SMA (ab5694, Abcam, 1:100 dilution), Collagne-1 (ab34710, Abcam, 1:800 dilution), and Fibronectin (F3648, Sigma, 1:1000 dilution). Briefly, deparaffinized and rehydrated sections were boiled in citrate antigen retrieval solution for antigen retrieval. After incubation with 10% goat serum, sections were incubated with primary antibodies overnight at 4 °C, followed by incubation with horseradish peroxidase-conjugated secondary antibodies (ZLI-9018, ZSGB-BIO, 1:20 dilution). DAB-positive staining was assessed by two experienced pathologists in a blinded manner and evaluated by $H$ score as described previously[40]. For immunofluorescence, deparaffinized and rehydrated sections were blocked and incubated with antibodies against KIM-1 (AF1817, R&D System, 1:200 dilution) overnight at 4 °C, followed by incubation with Alexa Fluor 555-conjugated secondary antibodies (Invitrogen, 2273776, 1:1000 dilution) and fluorescein-LTL (Vector Lab, FL-1321, 1:1000 dilution). The tissue slides were imaged with a fluorescence microscope (DMi8, Leica).

### Isolation of proximal tubules

Proximal tubules were isolated as described previously[41,42] with several modifications. Briefly, mice were perfused through the thoracic aorta with 20 mL of a cold magnetic bead solution supplemented with 40 μL of Dynabeads M450 (14013, Invitrogen). Kidney cortex was collected and minced into 1-mm³ pieces, which were then digested in HBSS supplemented with 1 mg/mL collagenase I (LS004196, Worthington), 0.75 mg/mL trypsin inhibitor (T6522, Sigma), and 40 U/mL DNase I (D4513, Sigma) at 37 °C for 10 min on a rotator. This digestion step was performed twice and the suspension was passed through cell strainers from 100 to 45 μm to remove large cellular debris. Digested tissues on the cell strainer were collected with cold PBS, and washed three times in cold PBS. The filtrates were placed on a Magnetic Separation Stand (Promega) to remove glomeruli. The long segments of proximal tubules in suspension were used immediately in subsequent experiments.

### RNA isolation and RT-qPCR

Total RNA was extracted from either isolated proximal tubules or whole kidney tissue with TRIzol (15596018, Invitrogen). Two micrograms of total RNA were reverse-transcribed using a cDNA Synthesis Kit (4897030001, Roche) according to the manufacturer's protocol. SYBR-based real-time quantitative polymerase chain reaction (RT-qPCR) was performed in a 96-well reaction plate to detect mRNA expression of *Rxra*, *Havcr1*, *Lcn2*, *Col3a1*, and *Fn1*. The primer sequences are presented in Supplementary Table S2.

### ATAC-seq and data analysis

Cell viability (>90%) was determined by trypan blue staining. A total of 100,000 cells were lysed in lysis buffer (10 mM Tris-HCl pH 7.4, 3 mM MgCl₂, 0.1% (v/v) IGPAL CA-630, 10 mM NaCl) for 20 min on ice. The nuclei were subjected to transposition with Tn5 transposase (TD501-01, Vazyme) for 30 min at 37 °C. After tagmentation, DNA was purified with VAHTS DNA Clean Beads (N41101, Vazyme) for final library construction and high-throughput sequencing on the Illumina NovaSeq platform.

FASTQ files were trimmed with trimGalore (v1.18), and the filtered reads were aligned to mouse reference genome (GRCm38/

mm10) with Bowtie2 (v2.3.5.1). Reads mapped to mitochondrial DNA were removed by removeChrom. Peak calling was performed using MACS2 (v2.2.7.1). Correlation analysis was performed using deepTools (v3.4.3). ATAC-seq differential peaks between tissue pairs were identified using HOMER with $P$ value < 0.01 as cutoff. HOMER was also used for motif enrichment analysis. For dynamic ATAC peak identification among different stages, we conducted a soft clustering analysis with the Mfuzz R package. Chromatin accessibility data showing a pertinence membership > 51% at a given stage were grouped into four Mfuzz clusters. ATAC signals were visualized as a heatmap using deepTools (v3.4.3). Sequencing data were deposited in the Gene Expression Omnibus (GEO) (GSE197815) and are available on the web (https://www.ncbi.nlm.nih.gov/geo/query/acc.cgi?acc=GSE197815).

### ChIP-seq and data analysis

ChIP was performed as previously described[40], with modifications. Isolated primary proximal tubules were crosslinked with 1% formaldehyde for 15 min at room temperature (RT). Crosslinking was stopped by glycine for 5 min at RT. Cells were lysed using cell lysis buffer (140 mM NaCl, 10% glycerol, 0.5% NP-40, 0.25% Triton X-100, 1 mM EDTA, 50 mM Tris-HCl pH 8.0), and then nuclei were isolated using nuclei lysis buffer (10 mM Tris pH 8.0, 1 mM EDTA, 0.5 mM EGTA, 0.2% SDS). Nuclei were sonicated to generate fragments in the range of 200–500 bp, which were incubated with antibody overnight at 4 °C (RXRα: ET7108-99, Huabio, 2 μg; H3K27ac: ab4729, Abcam, 2 μg). After elution and reverse crosslinking, the DNA fragments were used for library amplification with a VAHTS Universal Library Prep Kit (ND601, Vazyme) according to the manufacturer's instructions.

ChIP-seq data were analyzed as previously described[40]. In brief, sequencing was performed on the Illumina NovaSeq platform. FastQC (v0.11.9) software was used for quality control. Trimmed clean reads were aligned to the mouse reference genome (GRCm38/mm10) with Bowtie (v2.3.5.1). MACS2 (v2.2.7.1) was used for ChIP-seq peak calling and BED file generation. BigWig files were derived from deepTools (v3.4.3). Differential accessible regions were analyzed on normalized trimmed counts using HOMER with $P$ value < 0.01 as cutoff. Annotation of peaks was performed using HOMER. Genomic data were deposited in the GEO (GSE197815) and are available on the web (https://www.ncbi.nlm.nih.gov/geo/query/acc.cgi?acc=GSE197815).

### RNA-seq and data analysis

Total RNA was isolated from primary proximal tubules using TRIzol and RNA quality was assessed using an Agilent 2100 Bioanalyzer. Total mRNA was used for RNA-seq library construction. Sequencing was performed on the Illumina NovaSeq platform. Adaptor-trimmed RNA-seq reads were aligned to the mouse reference genome (GRCm38/mm10) with HISAT2 (v2.1.0). Mapped reads were quantified using featureCounts (v1.6.0). Differential expression was calculated with the DESeq2 package[43]. Differentially expressed genes (DEGs) were determined using a cutoff of >1.5-fold change with adjusted $P < 0.05$. Sequencing data were deposited in the GEO (GSE197815) and are available on the web.

### Public scRNA-seq data analysis

Raw mouse AKI scRNA-seq counts and meta data were downloaded from GEO (GSE139107), while pre-processed and integrated human AKI scRNA-seq data in h5seurat format were accessed through the Kidney Precision Medicine Project (KPMP) biorepository website at https://atlas.kpmp.org/repository/ after approval by KPMP. The PTCs previously defined in the original research, were filtered for downstream analysis through the standard pipeline of Seurat (V4)[44]. In detail, for the mouse data, the raw count matrix was normalized and scaled when the effect of UMI and mitochondrial percentage were removed by setting the parameter "vars.to.regress". The union of DEGs of the PT cell injury states provided by the original research was selected as highly variable genes (HVGs) for principal component analysis (PCA). Time series analysis of PT cells was performed in the unintegrated UMAP (uniform manifold approximation and projection) space reduced upon the first six principal components. Unsupervised clustering with resolution 0.4 and subsequent annotation with injury state markers displayed in the original research revolved eight clusters defined by associated states: healthy PT segment 1 (H_S1), healthy PT segment 2 (H_S2), healthy PT segment 3 (H_S3), mild injury PT segment 1 (MI_S1), mild injury PT segment 2/3 (MI_S23), severe injury (SI), repairing (R), and failed repair (FR). H_S1, H_S2, and H_S3 were merged as healthy (H), and MI_S1 and MI_S23 were merged as mild injury (MI). The RXRα-activated genes were assembled as a gene set for input into the "AddModuleScore" function to calculate the enrichment score of RXRα-mediated genes. Visualization of $Rxra$ or RXRα-activated genes by heatmaps or violin plots was performed by the functions "pheatmap" in the pheatmap package and "Vlnplot" in the Seurat package. We then constructed a developmental trajectory model of the injury severity of PT cells with the standard Monocle3 pipeline[45] to investigate the correlation between $Rxra$ and RXRα-activated genes or PT injury severity. The pseudo-time of each PT cell was calculated and used to align the cells. As pseudo-time increased, state label demonstrated a tendency from H to MI to SI, and state-associated markers gradually peaked in the consistent state area, suggesting a correlation between pseudo-time and injury severity and therefore successful model construction. Visualization of $Rxra$ or RXRα-activated genes in a heatmap with aligned pseudo-time was performed by "plot_pseudotime_heatmap". A similar protocol was followed for the human data, except that the deeper denoising step with normalization and scaling by sctransform were performed and followed by integration with harmony to remove the batch effect of the specimen[46,47]. We annotated each human PT cell using transcriptomics similarity analysis and comparing with the previously processed reference mouse data[16]. In this step, the objective annotation tool scMCA was used to determine the objective injury state of human PT cells[48,49].

### Statistics

GraphPad Prism 8.0 was used for statistical analysis. Student's $t$-test or Mann–Whitney $U$ test was used to evaluate the difference between two groups depending on the data distribution. The Correlation test was performed for correlation analysis between two variables. A $P$ value below 0.05 was considered significant. The results are presented as means ± SEM. DEGs were identified by DESeq algorithms[43]. Adjusted $P$-values (FDR) were calculated according to the Benjamini–Hochberg rule. The test performed is the Wald test, which is a test for coefficients in a regression model.

### Reporting summary

Further information on research design is available in the Nature Portfolio Reporting Summary linked to this article.

## Data availability

The main data supporting the findings of this study are available within this Article, it's Supplementary Information and Source Data. All ATAC-seq, ChIP-seq, and RNA-seq raw data files are available in the Gene Expression Omnibus (GEO) with the accession number GSE197815. ATAC-Seq, ChIP-seq, and RNA-seq reads were aligned to the mouse reference genome (GRCm38/mm10). All other data can be found within the main manuscript or the source data file. Source data are provided with this paper.

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

## Acknowledgements

The authors thank KPMP for their assistance in the study. This work was supported by grants from the National Natural Science Foundation of China (92068113 and 32070634 to L.Z., 82070689 to Yu.C., and 82130021 to L.Y.), the National Key Research and Development Program of China (2017YFA0504102 to Yu.C.), the Natural Science Foundation of Tianjin (19JCJQJC63800 to Yu.C., 21JCJQJC00100 to L.Z.), Tianjin Municipal Education Commission (2019ZD027 to Yu.C.), the Beijing Young Scientist Program (BJJWZYJH01201910001006 to L.Y.), Nature and Science Foundation of Guangdong province (2019B1515120075 to J.N.), and Youth Fund of the Second Hospital of Tianjin Medical University (2021ydey01 to X.C.).

## Author contributions

X.C. and J.W. performed animal and biochemistry studies. X.C. and Z.Li. performed bioinformatic analysis. T.Z. performed animal studies. Z.Liu., Yi.C. and Q.Y. assisted with the biochemical studies. L.Y. provided human specimens. Y.Z. performed human specimen analysis. T.L., J.N., and Y.N. edited the manuscript. X.C. and L.L. analyzed data and wrote the manuscript. L.Y. conceived and designed the project and edited the manuscript. L.Z. and Yu.C. conceived and supervised the project, analyzed data, and wrote the manuscript.

## Competing interests

The authors declare no competing interests.
