## [Peer Review File · Nature Communications]

Chromatin Accessibility Dynamics Dictate Renal Tubular Epithelial Cell Response to InjuryREVIEWER COMMENTS

Reviewer #1 (Remarks to the Author):

Cao et al present a manuscript studying the chromatin dynamics of mouse tubular epithelial cells (TECs) in response to mild and severe ischemia reperfusion injury (MI, SI) induced by temporary renal artery clamping. After MI, mouse kidneys are able to recover but after SI, maladaptive fibrosis and parenchymal loss develops. They isolate TECs from mouse kidneys at baseline and 2, 7 and 30 days after the actual or sham procedure. On these cells they generate ATAC-seq and RNA-seq. From analyzing the MI and SI groups, they identify a key role for the transcription factor RXRa in adaptive repair. Using ChIP-seq for RXRa, they show that it is bound to ~25% of the differentially accessible chromatin regions in TECs from MI mice. These regions are located close to important functional genes for TECs such as various transporters. RXRa is also lost in SI mice, perhaps explaining their inability to repair adaptively. Next, they treat their SI mice with an RXRa agonist, bexarotene (Bex) and see improvement of kidney function, histology and multi-omic signatures towards adaptive repair. They then cross-validate their findings in scRNA-seq data from KPMP and demonstrate loss of RXRa in patient biopsies with clinical and histologic AKI.

The experiments are well designed and the analyses well executed. These data represent a valuable contribution to the field. Some suggestions for improvement are listed below.

Comments:

Comparison to the mouse models of AKI and repair (Kirita et al, PNAS 2020; Dhillon et al, Cell Metabolism 2021) is well done. However, the authors should also compare their findings to Lidberg et al, JASN 2022 (PMID: 35197326). In that paper, the authors show that normal human kidney cortex has enrichment of RXRa motifs in open chromatin regions, which is diminished in injury. This would seem to support the authors' conclusions in a human system.

The cause of the AKI in patients should be listed in Supplementary Table 1. Also, classification of patients by histology alone is problematic since pre-analytic variable (e.g. time between biopsy procurement to stabilization) may impact apparent AKI histology. Also in Suppl. Table 1, the degree of SCr elevation doesn't always match with histologic severity. To explore this, the authors could explore how does the clinical classification of the patient's AKI (e.g. KDIGO AKI Stage 1, 2 or 3) correlates with RXRa expression in the tissue? What happened to the patients? Did they recover or have persistent loss of kidney function? Did recovery inversely correlate with degree of RXRa loss? This has implications for whether RXRa staining in biopsies may be helpful in identifying patients who might repair adaptively or progress to fibrosis.

If RXRa is lost in the murine model of SI, then what is Bex binding to and how is it improving kidney function? Or is this a matter of degree of RXRa loss, not complete loss?

In the statistics section of the methods, multiple testing correction in analyses is not explicitly stated and should be included.

The abstract doesn't state that they are using a mouse model of ischemia reperfusion injury - perhaps this is Nat. Comms' style, but I think the model should be included.

Immunohistochemistry panels in Fig. 5f and 8f could have better brightness/contrast to make the brown signal more apparent

For figures 5f, 6f, and 8f, the figure legend states "Immunofluorescence staining of ...". This should be "immunohistochemistry". Please verify this throughout the manuscript as well.

Reviewer #2 (Remarks to the Author):

The results are interesting and well done from a technical perspective although the comparisons of SI and MI are influenced by the marked difference in mortality in the two models. There is also concern that inflammatory cells may also be isolated especially in the surviving SI group and this may influence the conclusion regarding whether the chromatin changes are exclusively from epithelial cells.

1. Suppl Fig 1: Would be useful to include protein levels of fibronectin and col3a1 in addition to qRT-PCR. Also it would be important to report on mortality rates in the two ischemic models since mortality in the severe ischemia group would affect the comparisons by biasing toward less injury. This is particularly important since there are only 3 animals for each condition, at least as reflected for Havcr1, Lcn2, Col3a1 and Fn1. Given that Fig 6a indicates only 30% survival in the SI group it must be concluded that the findings throughout the manuscript are very significantly biased by survival bias. One can only analyze cells from SI at day 7 if the animals survive to day 7. Since 70% of animals do not how can we rely on the reported differences between day 2 and day 7 in MI and SI?

Would also suggest supplementing isolated TEC data in 1c with whole tissue staining and RT-PCR. Statistical differences should be included in Suppl Fig 1b. The authors state in the text that this figure demonstrates functional recovery with shorter ischemia times yet there are no differences in creatinine at 30 days between the MI and SI conditions. There are clearly changes in Col3a1 and Fn1 by RT-PCR. If these differences are confirmed at the protein level then they should modify their conclusion (page 6, line 19) to reflect these differences but the authors have not measured function at 30 days.

2. Using the technique for proximal tubule isolation it does not appear that cortical distal tubules or collecting duct elements were removed. It might be more appropriate to call them tubule epithelial cells that are predominantly proximal tubules. In fact they often refer to the cells at TECs in text but describe them at PTCs in the Figures.

In addition in the post-ischemic state there is inflammation and increased numbers of inflammatory cells. These would be expected to be present in the cortical cells isolated. Therefore when the authors conclude that on day 7 there are inflammatory genes activated they cannot conclude that these are in proximal tubule cells. In fact one would expect more inflammation in the severe injury group. The representation in Fig 2a that the analyses are done on PTCs does not recognize the contribution of inflammatory cells (e.g. macrophages and T-cells and possibly neutrophils).

3. In Fig 3a there are many more MI-DARs on day 7 than there are SI-DARs. Can the authors be sure that the increased tubular injury seen in the SI-DARs does not result in more loss of cells after isolation procedures?

4. On page 9 (lines 11-18) after describing the numbers of MI and SI-DARs at 2 days and 7 days the authors conclude that MI TECs and SI TECs exhibit distinct chromatin accessibility landscapes on both day 2 and 7; yet with data presented in 3a alone they cannot conclude this for day 2 at this point in the results section.

5. Page 12, lines 10 and 11. The authors state that the involvement of RXR α has not been reported in AKI previously. In fact RXR α increased expression has been described previously in models of AKI (e.g. Elsayed et al. Naunyn-Schmiedeberg's Arch Pharmacol. 2016 389:327-37; Chiba et al. J Am Soc Neph 27:495-508, 2016). It is interesting that the latter reference suggests that retinoic acid signaling coordinates macrophage-dependent injury and repair after AKI and hence is directly relevant to this current manuscript. Retinoic acid signaling has an important role in kidney development and hence it is expected to play a role in kidney repair.

6. Bexarotene shows a significant effect by 2 days and hence longer term effects can be interpreted as reflecting this effect on injury and survival. Activation of the retinoid system has been associated with protection against brain ischemic injury and has been implicated with many other associated effects such as on autophagy, bcl2 and other signaling consequences. The most relevant effector consequences of bexarotene to protect against the early injury are not explored in this manuscript

7. In the Discussion the authors comment that they have discovered a series of stress-related TFs, including bZIP, TEAD, and ETS, that may function as regulators controlling the openness of chromatin during severe kidney injury. There is very little information implicating these transcription factors as important contributors to repair or lack thereof in the paper. bZIP family members have been associated with AKI previously (e.g. Cheng and Lin, *Tox Mech Methods* 2011 21: 362-6, Yan et al *Ann Med* 2018. 50:381-390). ETS has been reported to influence recovery before (Tanaka et al *J Am Soc Nephrol* 2004, 15:3083-92)

Reviewer #3 (Remarks to the Author):

The authors profiled active DNA regulatory elements by ATAC-seq to identify the loci responsible to protection from ischemia-induced mouse kidney injury by comparing mild injury (MI) and severe injury (SI). By integrating with RNA-seq results, they identified RXRa as a key transcription factor in promoting adaptive repair from SI. Expression of RXRa is reduced in the kidneys of severe mouse and human AKI, and is positively correlated with kidney function. Activation of RXRa by bexarotene, an RXRa agonist protects TECs against severe kidney injury through restoring the chromatin state and gene expression.

The findings are very interesting. However, some major concerns should be addressed.

Major concerns

According to Figure 4, RXRA is not a very strong candidate. HNF4A and PPARA are much better candidates. HNF4A, PPARA and RXRA are reported in multiple disease models, such as glomerulosclerosis, renal fibrosis, and acute kidney injury. Therefore, the authors couldn't find new targets by ATAC-seq strategy. One recent paper (<https://www.jci.org/articles/view/140155>) reported adverse effects of retinoic acid signaling in kidney. HNF4A or PPARA may be better to treat this mouse model of kidney injury?

Only 25 % of MI-DAR overlapped with RXRA according to Fig.5a. I think authors should have studied with HNF4A and PPARA. At least they should show overlap of MI-DAR with HNF4A or PPARA in parallel (can be positive controls).

Source of human healthy cases is not clear. Are they healthy parts of kidney cancer? Or how did they get biopsies from healthy individuals?

I couldn't find institutional approval of these experiments (IRB and IUCUC for animal experiments and human samples tissues). Are they really approved by their Institute?

According to IGV tracks, ATAC-peaks are not always at the promoter regions. Are they really important for the expression of the genes (especially Fig.2e)? If they did GO analysis based on those peaks, is it reliable? Are differences in the heights of ATAC-peaks statistically significant between samples? Also in Fig.1e, 3g, 5i, 7f.

Title is "Chromatin Accessibility Dynamics Dictate Renal Tubular Epithelial Cell Response to Injury". It is very broad title and not clear what are the major findings of this manuscript.

In this manuscript, they just found the open loci by ATAC-seq and correlation with RXRA. To make

Chromatin Accessibility Dynamics clear, H3K27ac-ChIP seq should be done and confirm with the position of ATAC-peaks.

Regarding the opening chromatin, what nucleosome remodelers are involved with RXRA in this kidney injury model?

Fig.6

Dose of bexarotene. (100mg/kg/day). How did they decide the dose? At the first attempt, they need to do dose response and time course study including toxicity assay in mouse to determine the best dose and timing. Such data should also be shown.

Overall, descriptions are not enough in figure legends. Add necessary info as many as possible. Spell out abbreviations. Include numbers of samples, mice, and patients.

Fig.3. Sham samples should be included to compare with MI and SI. MI is similar to sham?

IHC of RXRA is not clear (Fig.5f and 8f).

Fig.8a. Consider to show the data in different (more meaningful) way.

Fig.8g. Is it possible to include Healthy and MI cases here?

REVIEWER COMMENTS

Reviewer #1 (Remarks to the Author):

Cao et al present a manuscript studying the chromatin dynamics of mouse tubular epithelial cells (TECs) in response to mild and severe ischemia reperfusion injury (MI, SI) induced by temporary renal artery clamping. After MI, mouse kidneys are able to recover but after SI, maladaptive fibrosis and parenchymal loss develops. They isolate TECs from mouse kidneys at baseline and 2, 7 and 30 days after the actual or sham procedure. On these cells they generate ATAC-seq and RNA-seq. From analyzing the MI and SI groups, they identify a key role for the transcription factor RXR α in adaptive repair. Using ChIP-seq for RXR α , they show that it is bound to ~25% of the differentially accessible chromatin regions in TECs from MI mice. These regions are located close to important functional genes for TECs such as various transporters. RXR α is also lost in SI mice, perhaps explaining their inability to repair adaptively. Next, they treat their SI mice with an RXR α agonist, bexarotene (Bex) and see improvement of kidney function, histology and multi-omic signatures towards adaptive repair. They then cross-validate their findings in scRNA-seq data from KPMP and demonstrate loss of RXR α in patient biopsies with clinical and histologic AKI.

The experiments are well designed and the analyses well executed. These data represent a valuable contribution to the field. Some suggestions for improvement are listed below.

Response: We thank the reviewer for this positive appraisal of our work.

Comments:

1. Comparison to the mouse models of AKI and repair (Kirita et al, PNAS 2020; Dhillon et al, Cell Metabolism 2021) is well done. However, the authors should also compare their findings to Lidberg et al, JASN 2022 (PMID: 35197326). In that paper, the authors show that normal human kidney cortex has enrichment of RXR α motifs in open chromatin regions, which is diminished in injury. This would seem to support the authors' conclusions in a human system.

Response: We thank the reviewer for this suggestion and have accordingly re-analyzed the chromatin accessibility profile data sets generated by Lidberg and colleagues. We first identified kidney cortex-specific accessible regions and serum culture-specific accessible regions (Figure **a**). By quantifying the coverage of TF-binding motifs, RXR α motifs were found in 17.5% of kidney cortex-specific accessible regions (Figure **b**). Furthermore, the RXR α -bound regions were more accessible than those without RXR α binding, and the mRNA levels of RXR α -bound genes were higher than those of RXR α -unbound genes (Figure **c** and **d**). These results indicate that RXR α binding correlates positively with both chromatin openness and gene activation in normal human kidney cortex.

To compare our findings with these human data, we then identified the RXR α -bound DARs and compared them between sham and either MI or SI at day 2. As shown in Figure **e** and **f**, compared with MI or SI, RXR α -bound normal mouse TEC-specific regions significantly overlapped with

human normal kidney cortex. These results provide human relevance for our studies, and we have therefore cited this article to support our proposal that RXR also plays a role in human renal injury models [Page 20, Lines 376-377 & Page 21, line 378]. However, the injury model in Lidberg et al.'s study is largely different from ours, and we identified RXR α by comparing MI-DARs and SI-DARs. Thus, to avoid the possibility of confusing readers, we chose not to present these data in the present study.

Binding of RXR α in normal kidney cortex. (a) Heatmap visualization of kidney cortex-specific accessible regions and serum culture-specific accessible regions. (b) Occupancy of RXR α on kidney cortex-specific accessible regions. (c) Signal profile of RXR α ^{bound} and RXR α ^{unbound} sites. (d) Gene expression levels of RXR α ^{bound} and RXR α ^{unbound} site-associated genes. (e and f) Occupancy of RXR α on normal mouse TEC-specific regions (left). Venn diagram showing the overlap of RXR α ^{bound} genes in normal kidney cortex and normal TECs (right).

2. The cause of the AKI in patients should be listed in Supplementary Table 1. Also, classification of patients by histology alone is problematic since pre-analytic variable (e.g. time between biopsy procurement to stabilization) may impact apparent AKI histology. Also in Suppl. Table 1, the degree of SCr elevation doesn't always match with histologic severity. To explore this, the authors could explore how does the clinical classification of the patient's AKI (e.g. KDIGO AKI Stage 1, 2 or 3) correlates with RXR α expression in the tissue? What happened to the patients? Did they recover or have persistent loss of kidney function? Did recovery inversely correlate with degree of RXR α loss? This has implications for whether RXR α staining in biopsies may be helpful in identifying patients who might repair adaptively or progress to fibrosis.

Response: We thank the reviewer for these insightful questions and suggestions.

All of the patient data are re-organized in the new version of Supplementary Table 1. We added a further 11 patients with pathologically diagnosed acute tubular injury who were treated in Peking University First Hospital. The clinical data for these 28 patients (including the 17 patients in the original manuscript) were supplemented with information about the etiology of the AKI, serum creatinine (Scr) at biopsy, BUN at renal biopsy, serum creatinine at peak, AKI/AKD stage according to the KDIGO criteria. (Page 18, Lines 341-345) We also included the degree of renal tubular injury,

assessed by two renal pathologists according to a 0 to 4+ scale scoring system.

We examined the relationship between RXR α expression level and the severity of acute tubular injury, and the clinical parameters of serum creatinine at biopsy, BUN at biopsy, serum creatinine at peak, AKI/AKD peak stages, and renal prognosis at discharge.

RXR α expression decreased substantially in kidneys from patients with severe acute tubular injury (new Fig. 8f). More importantly, RXR α expression was lower in patients at AKI stages 2 and 3 than in patients at AKI stage 1, indicating that RXR α expression is inversely correlated with the clinical severity of AKI (new Fig. 8g). Further analysis revealed that RXR α expression level was also inversely correlated with the serum creatinine (Scr) at peak, BUN at peak, and Scr at biopsy (new Fig. 8h). (Page 19, Lines 350-354) We also tried our best to collect information on the patients' renal function outcome. Unfortunately, we were unable to obtain pre-discharge creatinine for nine patients; and one patient was categorized as Non-AKI or Non-AKD according to the KDIGO criteria. We could not assess the renal function recovery for these 10 patients. Among the remaining 18, we compared each patient's renal function outcome with the expression of RXR α . As shown in the following figure, we found no significant difference between the recovery and non-recovery groups ($P = 0.1342$). This may be due to the small patient number in the non-recovery group; it may also be related to the complex inducing factors of AKI in these patients and/or interference from the various therapeutic methods used to treat them. Further analysis is therefore needed with a larger number of patients.

3. *If RXR α is lost in the murine model of SI, then what is Bex binding to and how is it improving kidney function? Or is this a matter of degree of RXR α loss, not complete loss?*

Response: Yes, it is a matter of degree. As shown in Fig. 5f, the expression level of RXR α decreased, but could still be readily detected in the SI mouse kidney. The mechanism of Bex action is to bind the RXR α ligand-binding domain, which alters RXR α conformation and promotes its binding with other partner proteins; these heterodimer protein complexes then bind to the promoters of target genes and activate gene expression. Therefore, even though the expression of RXR α decreases, Bex can still enhance target gene expression by activating residual RXR α protein.

4. *In the statistics section of the methods, multiple testing correction in analyses is not explicitly stated and should be included.*

Response: We thank the reviewer for pointing this out and have accordingly included more information about statistical analysis. Differentially expressed genes were identified by DESeq2 [PMID: 25516281] algorithms. Adjusted *P*-values (FDR) were calculated according to the Benjamini–Hochberg rule. The test performed is the Wald test, which is a test for coefficients in a regression model. (Page 32, Lines 626-628 & Page 33, Lines 629)

5. *The abstract doesn't state that they are using a mouse model of ischemia reperfusion injury - perhaps this is Nat. Comms' style, but I think the model should be included.*

Response: We agree with this suggestion and have therefore added this information in the Abstract. (Page 3, Lines 40-41)

6. *Immunohistochemistry panels in Fig. 5f and 8f could have better brightness/contrast to make the brown signal more apparent.*

Response: We have replaced these original images with higher-resolution and better brightness/contrast images.

7. *For figures 5f, 6f, and 8f, the figure legend states "Immunofluorescence staining of ...". This should be "immunohistochemistry". Please verify this throughout the manuscript as well.*

Response: We thank the reviewer for pointing this out and have revised the text accordingly.

Reviewer #2 (Remarks to the Author):

The results are interesting and well done from a technical perspective although the comparisons of SI and MI are influenced by the marked difference in mortality in the two models. There is also concern that inflammatory cells may also be isolated especially in the surviving SI group and this may influence the conclusion regarding whether the chromatin changes are exclusively from epithelial cells.

1. *Suppl Fig 1: Would be useful to include protein levels of fibronectin and col3a1 in addition to qRT-PCR. Also it would be important to report on mortality rates in the two ischemic models since mortality in the severe ischemia group would affect the comparisons by biasing toward less injury. This is particularly important since there are only 3 animals for each condition, at least as reflected for Havcr1, Lcn2, Col3a1 and Fn1. Given that Fig 6a indicates only 30% survival in the SI group it must be concluded that the findings throughout the manuscript are very significantly biased by survival bias. One can only analyze cells from SI at day 7 if the animals survive to day 7. Since 70% of animals do not how can we rely on the reported differences between day 2 and day 7 in MI and SI?*

Response: We thank the reviewer for these suggestions, in response to which we examined the expression of Fibronectin and Collagen-1 in MI and SI models. As shown in new Supplementary

Fig. 1g, protein levels of fibronectin and collagen-1 were also substantially upregulated in the mouse kidneys of SI, while the expression of these fibrotic genes was not statistically different between the sham group and the MI group. These results were consistent with the RT-PCR analysis. (Page 7, Lines 115-118).

We are grateful to the reviewer for raising this critical concern regarding survival bias. As shown in the new Supplementary Fig. 2, all mice survived in the MI group. Consistent with the survival rate in Fig. 6a, 39.58% of mice survived in the SI group. Under SI, mice began to die two days after surgery. For the SI group, the only data we could collect on day 7 and 30 were from mice that survived. Therefore, there is indeed an issue of this inevitable survival bias. However, compared with the MI group, the surviving mice in the SI group all underwent the transition from AKI to CKD and developed fibrosis at day 30 (new Supplementary Fig. 1f and g). Thus, the difference between adaptive and maladaptive repair still can be addressed. (Page 8, Lines 124-127)

As shown in new Supplementary Fig. 2b, creatinine levels of mice in the SI group on day 2 showed individual differences. The higher the creatinine level, the more likely the mice were to die. Therefore, to minimize survival bias and increase data consistency, we chose mice with serum creatinine in the range of 2.12 ± 0.17 mg/dL in the SI group and of 1.3 ± 0.24 mg/dL in the MI group for ATAC-seq and RNA-seq analysis (new Supplementary Fig. 2c). Since the mice were sacrificed for subsequent sequencing analysis on day 2, we could not know whether particular mice would have survived. However, within the above creatinine ranges, mice in the SI group were more likely to survive. Importantly, the extent of the initial injury was in the same range and could therefore be used for subsequent comparisons between day 2 and day 7. We have added the above information in the Results. (Page 8, Lines 124-127)

2. Would also suggest supplementing isolated TEC data in 1c with whole tissue staining and RT-PCR. Statistical differences should be included in Suppl Fig 1b. The authors state in the text that this figure demonstrates functional recovery with shorter ischemia times yet there are no differences in creatinine at 30 days between the MI and SI conditions. There are clearly changes in Col3a1 and Fn1 by RT-PCR. If these differences are confirmed at the protein level then they should modify their conclusion (page 6, line 19) to reflect these differences but the authors have not measured function at 30 days.

Response: In accordance with the reviewer's suggestion, we performed immunofluorescence staining for KIM-1 in MI and SI mouse kidneys. As shown in new Supplementary Fig. 1e, compared with the sham surgery group, the expression of KIM-1 was markedly increased in TECs two days after injury, and the increase was more pronounced in the SI group than in the MI group. We also analyzed injured marker gene expression with whole kidney tissue. Consistent with the results from isolated TECs, expression of *Havcr1* and *Lcn2* both increased (new Supplementary Fig. 1c). (Page 7, Lines 108-113)

We did a statistical analysis for the data in Supplementary Fig. 1b. As shown in the new Supplementary Fig. 1b, creatinine and BUN levels were significantly higher in the SI group than in the MI group at all time points tested. At day 30, creatinine and BUN levels returned to normal in

the MI group but not in the SI group. Together with the confirmatory analysis of fibrotic genes at both mRNA and protein levels, these results support our conclusion that MI can resolve with adaptive repair of functional recovery, while SI induces maladaptive repair leading to fibrosis.

3. Using the technique for proximal tubule isolation it does not appear that cortical distal tubules or collecting duct elements were removed. It might be more appropriate to call them tubule epithelial cells that are predominantly proximal tubules. In fact they often refer to the cells at TECs in text but describe them at PTCs in the Figures.

Response: We thank the reviewer for pointing this out and have revised the figures and figure legends accordingly to unify the terminology.

In addition in the post-ischemic state there is inflammation and increased numbers of inflammatory cells. These would be expected to be present in the cortical cells isolated. Therefore when the authors conclude that on day 7 there are inflammatory genes activated they cannot conclude that these are in proximal tubule cells. In fact one would expect more inflammation in the severe injury group. The representation in Fig 2a that the analyses are done on PTCs does not recognize the contribution of inflammatory cells (e.g. macrophages and T-cells and possibly neutrophils).

Response: The genome-wide analyses were performed with isolated TECs, not whole kidney cortex. To purify proximal tubules, the digested cortex suspension was passed through cell strainers from 100 to 45 μm . We used 100 μm cell strainers to remove non-digested kidney pieces, and 45 μm cell strainers to remove scattered cells and cellular debris. The long segments of proximal tubules in suspension were used immediately in subsequent experiments. As shown in the following immunofluorescence staining analysis (Figure **a**), LTL-positive cells exceeded 90% for each condition and no differences were observed among sham, MI, and SI groups. We also measured expression of an immune cell marker gene (*Cd45*), a macrophage marker gene (*Cd68*), and a T-cell marker gene (*Dd3e*) in whole cortex and purified tubule fragments. As predicted by the reviewer, the expression of these immune cell marker genes was markedly increased after injury in whole cortex, with the highest expression in the SI group. These data indicate that inflammatory cells infiltrate after injury, especially in SI kidneys. In contrast, in purified tubular cells, the expression levels of these marker genes were very low and did not differ between the injured and control groups (Figure **b**). Taken together, these results suggest that tubular cells were isolated with high purity.

To confirm that the identified inflammatory genes from our ATAC-seq analysis are activated in TECs, we reanalyzed the recently published single cell dataset [PMID: 32571916]. We assembled the identified inflammatory genes as a gene set and calculated the enrichment score in mouse scRNA-seq. As shown in the following Figure **c**, these genes were also significantly enriched in SI tubular cells, indicating that they are indeed activated in TECs.

Purification of isolated TECs. (a) Immunofluorescence staining of LTL-positive cells. (b) qRT-PCR analysis of *Ptprc* (CD45), *Cd3e*, and *Cd68* expression in isolated TECs. (c) Gene set scores of identified inflammatory genes.

4. In Fig 3a there are many more MI-DARs on day 7 than there are SI-DARs. Can the authors be sure that the increased tubular injury seen in the SI-DARs does not result in more loss of cells after isolation procedures?

Response: Though more severe injury did lead to greater cell loss, we unified the number of cells used in sequencing analysis. As described in Methods, all ATAC-seq experiments were performed with the same number of cells (100,000 live cells analyzed by trypan blue staining). (Page 28, Lines 532-534)

5. On page 9 (lines 11-18) after describing the numbers of MI and SI-DARs at 2 days and 7 days the authors conclude that MI TECs and SI TECs exhibit distinct chromatin accessibility landscapes on both day 2 and 7; yet with data presented in 3a alone they cannot conclude this for day 2 at this point in the results section.

Response: In Figure 3a, we compared the differences in chromatin openness between MI and SI groups on day 2 and day 7. In comparison to the SI group, we identified 4172 gained accessible regions specific to the MI group on day 2, which were defined as MI-DARs. Likewise compared with the MI group, 4253 gained regions were identified in SI group, which were defined as SI-DARs. So there were a total of 8425 differential regions between MI and SI. On day 7, the numbers

of MI-DARs and SI-DARs were 6309 and 2616, respectively, making a total of 8925 differential regions between MI and SI. Thus, the total numbers of DARs are similar between day 2 and day 7. We have added the total numbers of DARs to make the conclusion clearer. (Page 11, Lines 185-188)

6. Page 12, lines 10 and 11. The authors state that the involvement of RXR α has not been reported in AKI previously. In fact RXR α increased expression has been described previously in models of AKI (e.g. Elsayed et al. Naunyn Schmiedeberg's Arch Pharmacol. 2016 389:327-37; Chiba et al. J Am Soc Neph 27:495-508,2016). It is interesting that the latter reference suggests that retinoic acid signaling coordinates macrophage-dependent injury and repair after AKI and hence is directly relevant to this current manuscript. Retinoic acid signaling has an important role in kidney development and hence it is expected to play a role in kidney repair.

Response: We thank the reviewer for pointing this out. Elsayed et al. investigated the effect of cisplatin treatment on RAR α and RXR α expression in rat kidney injury. In addition, they examined the possible modulatory effects of an RAR agonist, all-trans retinoic acid (ATRA), on cisplatin-induced nephrotoxicity. However, the mechanism and function of RXR α in ischemia–reperfusion kidney injury has not been described yet. We have therefore revised the text. (Page 14, Lines 253-255)

Regarding retinoic acid in kidney development and injury, previous studies have mainly focused on RAR α [PMID: 26109319]. The nuclear receptors downstream of the retinoic acid signal control different target genes by interacting with distinct partner proteins. Our genome-wide and bioinformatics analysis identified RXR α as a key regulator in ischemia–reperfusion injury. As bexarotene is a specific agonist of RXR α , our mechanistic and functional studies are focused on understanding the mechanism of action of RXR α . However, as commented by the reviewer, the role of RXR α in retinoic acid signaling-mediated renal protection, and the interaction between RXR α and RAR α or other partner proteins in AKI, are very worthy of future research.

7. Baxarotene shows a significant effect by 2 days and hence longer term effects can be interpreted as reflecting this effect on injury and survival. Activation of the retinoid system has been associated with protection against brain ischemic injury and has been implicated with many other associated effects such as on autophagy, bcl2 and other signaling consequences. The most relevant effector consequences of bexarotene to protect against the early injury are not explored in this manuscript.

Response: Yes, we agree with the reviewer. Although we have shown that Bex restores the chromatin state and gene expression program of TECs after SI, evidence that directly supports a protective role of Bex against early kidney injury through activation of RXR α is still lacking. In addition, whether other cellular processes and survival factors are involved in the renal protective effect of Bex remains to be explored. The use of tubular cell-specific RXR α knockout mice will help answer these questions. We have emphasized these unaddressed questions and provided prospects of future work in the Discussion. (Page 21, Lines 383-389).

8. In the Discussion the authors comment that they have discovered a series of stress-related TFs, including bZIP, TEAD, and ETS, that may function as regulators controlling the openness of

chromatin during severe kidney injury. There is very little information implicating these transcription factors as important contributors to repair or lack thereof in the paper. bZIP family members have been associated with AKI previously (e.g. Cheng and Lin, Tox Mech Methods 2011 21: 362-6, Yan et al Ann Med 2018. 50:381-390). ETS has been reported to influence recovery before (Tanaka et al J Am Soc Nephrol 2004, 15:3083-92)

Response: We thank the reviewer for providing the above key references supporting the potential roles of these identified stress-related TFs in AKI. Yes, more evidence is needed to verify the role of these transcription factors in AKI. We have therefore added this information and revised the text in Discussion. (Page 22, Lines 411-416)

Reviewer #3 (Remarks to the Author):

The authors profiled active DNA regulatory elements by ATAC-seq to identify the loci responsible to protection from ischemia-induced mouse kidney injury by comparing mild injury (MI) and severe injury (SI). By integrating with RNA-seq results, they identified RXR α as a key transcription factor in promoting adaptive repair from SI. Expression of RXR α is reduced in the kidneys of severe mouse and human AKI, and is positively correlated with kidney function. Activation of RXR α by bexarotene, an RXR α agonist protects TECs against severe kidney injury through restoring the chromatin state and gene expression.

The findings are very interesting. However, some major concerns should be addressed.

Response: We thank the reviewer for this positive appraisal of our work.

Major concerns

1. According to Figure 4, RXR α is not a very strong candidate. HNF4A and PPAR α are much better candidates. HNF4A, PPAR α and RXR α are reported in multiple disease models, such as glomerulosclerosis, renal fibrosis, and acute kidney injury. Therefore, the authors couldn't find new targets by ATAC-seq strategy. One recent paper (<https://www.jci.org/articles/view/140155>) reported adverse effects of retinoic acid signaling in kidney. HNF4A or PPAR α may be better to treat this mouse model of kidney injury?

2. Only 25 % of MI-DAR overlapped with RXR α according to Fig.5a. I think authors should have studied with HNF4A and PPAR α . At least they should show overlap of MI-DAR with HNF4A or PPAR α in parallel (can be positive controls).

Response: we would like to address the above two comments together. The reasons we chose RXR α are as follows:

1. In Figure 4a, we analyzed the enrichment of transcription factor (TF) motifs on MI- and SI-DARs. Among the nuclear receptor group TFs, RXR α ranked third, and HNF4 α and PPAR α ranked first and second, respectively. Since the importance of TFs in establishing and maintaining open chromatin also depends largely on other factors (including genome-wide

coverage of TF-binding motifs, TF expression in response to injury, and TF–TF interactions), further comprehensive bioinformatic analyses were performed. As shown in Figure 4d, RXR α was one of the most prominently enriched TF in the network among all the TFs after MI.

- We thank the reviewer for suggesting us to evaluate the overlapping of HNF4 α and PPAR α genome-wide distribution with MI-DAR. We have accordingly analyzed published HNF4 α [PMID: 32636391] and PPAR α [PMID: 31706703, 30975991, 32130908, and 28282965] ChIP-seq data sets. As shown in the following figure **a**, the occupancy of RXR α , HNF4 α , and PPAR α was detected in 24.95%, 10.18%, and 19.39% of MI-DARs, respectively. We further analyzed the overlapping of their occupancy in MI-DARs. As shown in the following figure **b**, while these three TFs have their own unique binding regions, they also co-localize with one or both of the others in many regions, suggesting that they interact with each other to establish MI-DARs cooperatively.
- As the reviewer mentioned, the mechanisms and functions of HNF4 α and PPAR α in kidney injury and repair have previously been extensively studied [PMID: 32636391 and 33301705]. Thus, our identification of these previously AKI-related TFs verifies the validity of our network analysis. In this study, besides understanding the role of chromatin dynamics in AKI, we would like to identify new factors involved in injury response and to develop new therapeutic strategies for AKI treatment. Since the roles and the mechanisms of action of RXR α in AKI are not clear, we therefore focused on this new target for further analysis.

(a) Occupancy of RXR α , HNF α , and PPAR α on MI-DARs. **(b)** Venn diagram showing overlap of RXR α , HNF α , and PPAR α binding sites.

Based on the above reasons, we chose RXR α as the candidate TF. However, as the reviewer suggested, other TFs, especially HNF4 α and PPAR α , may also play important roles in the establishment and maintenance of chromatin openness. Therefore, their roles and the underlying mechanisms are also worthy of further investigation. In particular, as we have proposed in the Discussion, whether these TFs play a synergistic role in activating the renal protective gene expression program and in protecting against AKI warrants future studies.

3. Source of human healthy cases is not clear. Are they healthy parts of kidney cancer? Or how did they get biopsies from healthy individuals?

I couldn't find institutional approval of these experiments (IRB and IUCUC for animal experiments and human samples tissues). Are they really approved by their Institute?

Response: Control tissues were obtained from macroscopically normal tissue adjacent to kidney cancer tissue in the same patient. Animal experiments and human studies were approved by Tianjin Medical University and Peking University First Hospital, respectively. The above information was included in Methods in the original manuscript. We have moved the Study approval text to the first section of Methods.

4. According to IGV tracks, ATAC-peaks are not always at the promoter regions. Are they really important for the expression of the genes (especially Fig.2e)? If they did GO analysis based on those peaks, is it reliable? Are differences in the heights of ATAC-peaks statistically significant between samples? Also in Fig.1e, 3g, 5i, 7f.

5. Title is “Chromatin Accessibility Dynamics Dictate Renal Tubular Epithelial Cell Response to Injury”. It is very broad title and not clear what are the major findings of this manuscript. In this manuscript, they just found the open loci by ATAC-seq and correlation with RXRA. To make Chromatin Accessibility Dynamics clear, H3K27ac-ChIP seq should be done and confirm with the position of ATAC-peaks.

Response: We would like to address the above two comments together. We appreciate the reviewer’s critical suggestion and have accordingly performed H3K27ac ChIP-seq analysis in purified TECs after sham surgery, MI, and SI. We analyzed H3K27ac peak signals in MI- and SI-DARs at day 2 and day 7. As shown in the new Figure 3d, the differential H3K27ac peak signals were higher at day 7 than at day 2 in both MI- and SI-DARs. These results were consistent with RNA-seq data and indicated that chromatin domains become accessible prior to active histone marks and gene expression. These data further support the importance of chromatin accessibility dynamics in AKI. (Page 12, Lines 207-212)

For ATAC peak identification in Figures 1 and 2, we conducted a soft clustering analysis with the Mfuzz R package. Chromatin accessibility data showing a pertinence membership > 50% at a given stage were grouped into four Mfuzz clusters. All the differential ATAC-seq peaks were identified using HOMER with $P < 0.01$ as the cutoff for statistical significance. (Page 29, Lines 542-547)

Yes, not all the ATAC-seq peaks appeared in promoter regions. As shown in the following figure (See next page), many of the ATAC-seq peaks are located in enhancer regions which also displayed H3K27ac active histone mark.

Genome browser view. Representative DARs at the indicated gene loci for TECs after mild injury (a) and severe injury (b).

6. Regarding the opening chromatin, what nucleosome remodelers are involved with RXRA in this kidney injury model?

Response: We currently don't know which chromatin remodeling complex(es) are involved. Biochemical analysis of RXR α -interacting proteins would provide clues for answering this question, which is worthy of future research.

7. Fig.6 Dose of bexarotene. (100mg/kg/day). How did they decide the dose? At the first attempt,

they need to do dose response and time course study including toxicity assay in mouse to determine the best dose and timing. Such data should also be shown.

Response:

Bex is an FDA-approved drug, and the pharmacokinetics and drug safety of oral administration are well described in rodents, dogs, and humans. [PMID: 11408365] A dose of Bex of 100 mg/kg/day has been widely used in many studies. [PMID: 22323736, 33108748, and 32202512] Wu et al. observed no obvious toxicity or weight loss in mice treated with Bex (100 mg/kg/day) daily for 7 days a week from the age of 6–8 weeks to the age of 7–8 months.

8. Overall, descriptions are not enough in figure legends. Add necessary info as many as possible. Spell out abbreviations. Include numbers of samples, mice, and patients.

Response: In accordance with the reviewer's request, we have added more of such descriptive information in the figure legends.

9. Fig.3. Sham samples should be included to compare with MI and SI. MI is similar to sham?

Response: As shown in the new Supplementary Fig. 4 the DARs in the MI group are similar to those in the sham group. (Page 11, Lines 185-188)

=

10. IHC of RXRA is not clear (Fig.5f and 8f).

Response: We have replaced these figures with higher-resolution images.

11. Fig.8a. Consider to show the data in different (more meaningful) way.

Response: We have replaced the heatmap with a dot blot.

12. Fig.8g. Is it possible to include Healthy and MI cases here?

Response: Here we studied the correlation between RXR α expression and kidney function specifically in AKI patients. A total of 28 patients (12 MI and 16 SI) were included for analyzing Scr at biopsy, BUN at biopsy, and Scr at peak (new Fig. 8h).

REVIEWERS' COMMENTS

Reviewer #1 (Remarks to the Author):

The authors have satisfactorily addressed all my concerns and the revised manuscript is substantially improved.

Reviewer #2 (Remarks to the Author):

The authors have very nicely addressed the concerns of the Reviewers.

Reviewer #3 (Remarks to the Author):

Authors have addressed concerns raised at the initial review.
Some figures shown in the response to reviewers' comments (but not shown in main or supplementary) may be included in supplementary info.

REVIEWERS' COMMENTS

Reviewer #1 (Remarks to the Author):

The authors have satisfactorily addressed all my concerns and the revised manuscript is substantially improved.

Response: We thank the reviewer for this positive appraisal of our revision.

Reviewer #2 (Remarks to the Author):

The authors have very nicely addressed the concerns of the Reviewers.

Response: We thank the reviewer for this positive appraisal of our revision.

Reviewer #3 (Remarks to the Author):

Authors have addressed concerns raised at the initial review.

Some figures shown in the response to reviewers' comments (but not shown in main or supplementary) may be included in supplementary info.

Response: We thank the reviewer for this suggestion. Data related to the purity of isolated TECs have been integrated into the main text. (Page 7, Lines: 120-131, Supplementary figure 3)